# On Inherent Reasoning of VLMs in Indoor Scene Layout Design

## Abstract

Large vision-language models (VLMs) such as GPT-4o, Llama-3.2 have shown remarkable capabilities in visual understanding and reasoning, prompting us to test their off-the-shelf ability to reason and act as a 3D design assistant. This study investigates VLMs' visual reasoning capabilities using 3D indoor scene layout synthesis i.e. placement of furniture in a room, as a test-bed. We study three key primitive abilities in this context: (1) communication of spatial locations, (2) reasoning about free space and object collision, and (3) reasoning about object alignment, orientation, and functionality, each crucial to creating a VLM agent-based scene layout synthesis pipeline. We evaluate five state-of-the-art VLMs, both proprietary and open, on a new dataset incorporating 3400 questions that assess VLMs' current visual reasoning abilities in our context. Our findings reveal several remarkable insights: (1) VLMs consistently prefer normalized coordinates for spatial communication over absolute coordinates or pointing with image markers. (2) Contrary to expectations, VLMs perform best with simplified sketch based scene representation or, most strikingly, with no visual input at all, compared to detailed renderings. (3) Free space reasoning remains challenging, with performance only slightly above random guessing, though frontier models show significant improvement with collision checking tools. Surprisingly, free space reasoning with clear visible collisions in the image can also fail. (4) Reasoning about object alignment, size, orientation and functionality together compounds errors leading to near chance performance on our dataset. These findings serve to offer insights into current potential and limitations of using VLMs off-the-shelf towards developing advanced visual assistants capable of understanding and manipulating 3D environments.

## 1 Introduction

Large vision-language models (VLMs) that take vision and text as input and output text such as GPT-4o, Claude, Gemini, Qwen-V2, Llama-3.2 have showcased unprecedented capabilities in generalized image understanding and reasoning. By bridging the gap between visual perception and linguistic expression, these models can interpret and describe complex visual scenes with remarkable accuracy, opening a path towards intelligent visual assistants or co-pilots that naturally interact with a human through text or speech.

With these models steadily becoming more capable and available, we set our sights on the north-star of visual assistants that work with 3D artists in creating and manipulating virtual 3D scenes for movies, gaming, virtual reality or simulation. Of course, understanding and interacting with 3D spaces goes beyond this goal and underpins fundamental capabilities of human cognition such as navigation, object manipulation, and spatial planning. For artificial agents to seamlessly integrate into either virtual or real 3D environments and assist in domains like 3D artistry, immersive virtual reality or robotics they must possess sophisticated 3D reasoning capabilities. As a first step towards virtual assistants for 3D design, we aim to evaluate off-the-shelf 3D reasoning abilities of pre-trained $\{vision, text\} \rightarrow \{text\}$ VLMs. As our test-bed, we choose indoor scene layout synthesis i.e. the placement of furniture within a given indoor room geometry, which is one of the many challenges on the path to a general 3D design visual assistant. This task demands understanding of individual objects, their spatial relationships, functional alignment and adherence to simple physical constraints such as collision avoidance through free space reasoning.

Recent studies, such as Feng et al. (2024); Yang et al. (2024); Çelen et al. (2024); Fu et al. (2024), have shown encouraging results using agent-based scene layout synthesis pipelines with off-the-shelf LLMs i.e. large language models without vision. These approaches show promising results, while their unique design choices lead to a lack of systematic understanding of the key underpinnings of 3D reasoning across the methods. For instance, Feng et al. (2024) implicitly combines 3D reasoning and action and utilize normalized coordinates to communicate 3D locations with the LLM. Yang et al. (2024); Çelen et al. (2024); Fu et al. (2024) each generate different text-based scene graph representations with spatial constraints using a language model and utilize a back-tracking based solver to convert constraints into a valid (if possible) 3D spatial arrangement of furniture in a room, leading to different facets of 3D reasoning coming from hard-coded solvers vs. the LLM itself. Beyond indoor layout synthesis and LLMs, Nasiriany et al. (2024) and Yang et al. (2023) have shown it is possible to communicate spatial locations with VLMs using text-based references to drawn markers on an image leading to enhanced open-world robotic planning and visual grounding when answering fine-grained visual questions respectively.

In this paper, instead of an indoor layout synthesis pipeline, we focus on systematically investigating primitive 3D reasoning i.e. perceiving and reasoning about object layouts, and acting i.e. placing objects at exact locations, capabilities essential towards creating purely VLM agent-based scene layout synthesis pipelines. Specifically, we decompose 3D reasoning abilities required in this context into three key primitives: (1) communication of spatial location/coordinates, (2) reasoning about free space and object collisions, and (3) joint reasoning about object alignment, orientation, and functionality. These primitives not only underpin scene layout synthesis, but also take first steps towards evaluating VLM-based agents as general 3D design assistants.

To evaluate these capabilities within the context of indoor scene layout design, we create a new evaluation dataset of indoor rooms adapting the 3D-FRONT dataset Fu et al. (2021). Our dataset comprises a total of 3400 questions across all three tasks and explores five different visual modalities, ranging from textual representations of the scene to simplified sketch based renderings and photo-realistic scene renderings. Taking inspiration from the literature, we also evaluate four distinct methods of communicating spatial locations using VLMs: absolute coordinates, normalized coordinates, visual markers and a combination of visual and textual markers.

We evaluate five state-of-the-art VLMs, including both proprietary and open-source models, and report several key findings in the context of our task:

- Preference for normalized coordinates: VLMs consistently prefer normalized coordinates for spatial communication over absolute coordinates or pointing with image markers.
- Effectiveness of simplified representations: Contrary to expectation, VLMs perform best with simplified sketch-based scene representations or, most strikingly, with no visual input at all, compared to detailed renderings.
- Challenges in free space reasoning: Free space reasoning remains challenging, with performance only slightly above random guessing. We find that the VLMs we test prefer to compute free-space mathematically, often leading to mistakes, even if a collision between objects is apparent in the image. Unsurprisingly, when provided with tools for collision checking, a frontier model like GPT-4o shows significant improvement.
- Compounded errors in complex reasoning: Visual reasoning about free-space, orientation, object alignment and functionality together compounds errors, leading to near-chance performance on our final task.
- Over-reliance on language processing: Models like GPT-4o, GPT-4o-mini, LLAVA-Next, and LLaMA-3.2 perform 10-20% worse when visual inputs are included. This suggests over-reliance on language understanding and indicates that current VLMs do not effectively utilize visual information for spatial reasoning tasks.

These findings highlight both the potential and limitations of using VLMs off-the-shelf in 3D reasoning tasks in the context of indoor scene layout design. Current VLMs often act blindly in the face of clear visual context required for the task, but also excel at symbolically breaking down a task, where computational errors lead to performance degradation. In a scenario where 3D scene information must only be grokked visually, they fail, even when prompted carefully. We hope this study helps inform the next generation of data used to train large VLMs to improve their capabilities towards becoming intelligent 3D design assistants.

## 2 RELATED WORK

We evaluate 3D reasoning in the context of indoor scene layout synthesis by testing three primitive skills necessary to build a VLM-agent based pipeline for the task, instead of proposing a novel pipeline similar to prior LLM-based pipelines (Feng et al., 2024; Yang et al., 2024; Çelen et al., 2024; Fu et al., 2024) discussed earlier.

**Spatial Understanding and Reasoning in VLMs.** Spatial reasoning in VLMs is primarily categorized into relative and quantitative approaches. Relative spatial reasoning has been extensively studied and distinguishes relationships between objects in 3D space using concepts such as "next to" or "in front of"(Agrawal et al., 2015; Johnson et al., 2016; Krishna et al., 2016; Suhr et al., 2018; Yang et al., 2019; Belz et al., 2018; Goyal et al., 2020; Majumdar et al., 2024), with some studies enhancing linguistic complexity and mitigating spatial biases (Liu et al., 2023; Kamath et al., 2023). Quantitative spatial reasoning involves estimating spatial attributes like sizes and distances from natural images without using external tools (Chen et al., 2024; Cheng et al., 2024; Liao et al., 2024b). Additionally, Tong et al. (2024a) explores 3D reasoning through multiple-choice questions focusing on depth order and relative distance. In this paper, we investigate primitive 3D reasoning about object layouts and actions such as placing objects at exact 3D locations, which are a step towards visual assistants aiding 3D artists in creating and manipulating virtual environments.

**Visual capabilities of VLMs.** Recent works have observed textual-bias in VLMs, often pronouncing them visually "blind." Tong et al. (2024b) identified "CLIP-blind pairs"—visually distinct images that CLIP Radford et al. (2021) perceives as similar—and found visual patterns that GPT-4V OpenAI (2023) struggles with due to inaccurate visual grounding. Similarly, Zhang et al. (2024) showed that some models struggle to understand mathematical diagrams, relying heavily on textual questions. Moreover, Wang et al. (2024a) created synthetic VQA tasks to evaluate models' abilities to navigate mazes and identify images in grid structures. Additionally, Rahmanzadehgervi et al. (2024b) demonstrated that VLMs consistently struggle with tasks requiring spatial information, such as identifying whether two circles overlap or which letter is being circled in a word, particularly when geometric primitives overlap or are close together. We take a similar focus to a new domain and study 3D reasoning through three key primitives with increasing levels of complexity, towards a vision of a capable 3D design assistant. We believe each of these works show new pathways towards data and tasks required to train the next generation of VLMs. Our evaluation includes novel sub-tasks and scene views that resemble those used in prior work on indoor synthesis or encountered in 3D modeling software, aimed at shedding light into 3D reasoning abilities and limitations of VLMs in the context of a valuable use-case of VLMs as 3D design assistants.

**VLMs for 3D understanding.** Various 3D-VLMs have been proposed that incorporate 3D data at inference time, adapting VLM architectures to process inputs like point clouds, depth maps, or multi-view images. These approaches explore model architecture design and require retraining or fine-tuning (Hong et al., 2023; Zhu et al., 2024; Cheng et al., 2024). Recently, (Deng et al., 2024) evaluated the sensitivity of 3D-VLMs to stylistic variations in semantically equivalent sentences, showing that these models struggle with such variations. We set our sights on the north-star of visual assistants that work with 3D artists in creating and manipulating virtual 3D scenes, and believe our exploration on the off-the-shelf capabilities of frontier VLMs such as GPT-4o and LLaMA-3.2-90B could be used to improve both frontier models, as well as fine-tuned 3D-VLMs.

**3D Vision Language Action models (VLAs)** VLAs are VLMs retrained or finetuned to output low-level actions (typically in robotics) as text tokens, often combaining large-scale VQA datasets with task-specific robotic data (Brohan et al., 2023; Kim et al., 2024). While most VLAs are trained on task-specific robotics data, PiVoT (Nasiriany et al., 2024) reinterprets robotic navigation as iterative visual question answering on off-the-shelf VLMs, demonstrating how VLMs can be prompted visually to elicit visual reasoning.

## 3 TASKS AND DATASET

In this section, we describe task and dataset construction for each of the three 3D reasoning primitives/tasks we evaluate VLMs on off-the-shelf. To remind the reader, we choose three primitive abilities necessary (but not sufficient) for a pure VLM agent-based pipeline for indoor scene layout synthesis, to serve as our test-bed: (1) communication of spatial locations, (2) reasoning about free

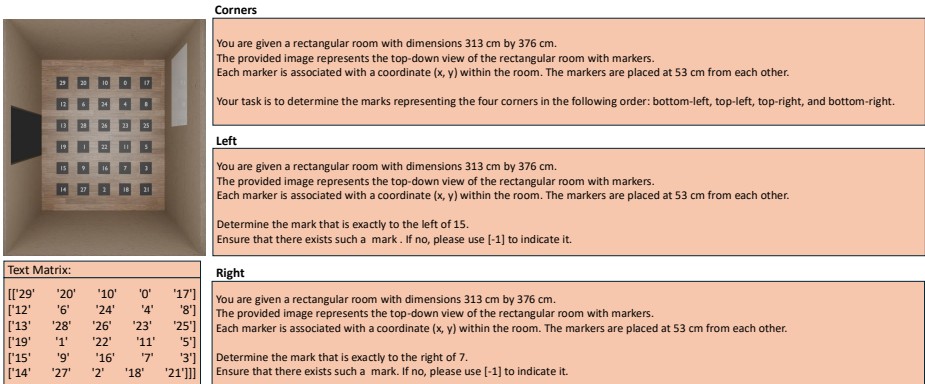

Figure 1: Example S1. communicating coordinates question showing 3D1 (top-down) rendering with markers to represent coordinates. We additionally use a text matrix representation of the marks when when no visual input is fed to the model or as additional redundant information (making OCR from the image unnecessary) along with visual markers.

space and object collisions, and (3) joint reasoning about free space, object alignment, orientation, and functionality. We refer to these three tasks as S1, S2 and S3 respectively in the following text. We construct our datasets using bedrooms from the 3D-Front dataset (Fu et al., 2021), which gives us realistic room sizes and object arrangements for various tasks.

We evaluate various state-of-the-art VLMs on these three tasks along two axes. Along the **visual axis**, we test across visual representations provided to the VLMs, illustrated in Fig. 3. This ranges from no visual input, where only textual descriptions represent the scene/room, to simple sketches consisting of a 2D top-down view of the scene/room, and extends to three types of 3D renderings created with Blender (Community, 2018): (i) *Top-Down View* (**3D1**), a common representation used in prior work; (ii) *Perspective View* (**3D2**), offering a non-axis-aligned perspective often encountered when working with 3D modeling software; and (iii) *Embodied View* (**3D3**), providing close to a first-person perspective that introduces complexity through occlusion and perspective distortion, but could potentially be closer to a distribution of natural images captured from a human perspective.

The second axis involves **methods to represent spatial coordinates**, which is also the focus of our first task. We explore (a) *Absolute Coordinates*, where we use realistic metric coordinates of rooms and objects; (b) *Normalized Coordinates*, assessing whether scaling positions to a standard range improves spatial reasoning, following prior work (Feng et al., 2024) and (c) *Discretized positions with spatial markers*, inspired by methods such as Pivot (Nasiriany et al. (2024) and Set-of-Marks Yang et al. (2023), where the model is allowed to refer to predefined discrete visual or textual markers assigned to specific positions instead of precise coordinates. We refer to this method as *marks* in subsequent text. Absolute and normalized coordinates are represented on our visual axes by rendering the coordinates of corners of the room (see Fig. 3. We use rectangular rooms only, following Layout-GPT (Feng et al., 2024)). For discretized spatial markers (marks), we use a regular grid of markers with constant separation rendered on the room floor (see Fig. 1 for an example). In the case of marks, we also experiment with a fourth approach providing the grid of marks as a 2D array as additional redundant information, which we refer to as *marks + text*.

For each task, visual and coordinate representation axis, we design VQA (Antol et al., 2015; Yue et al., 2024) style questions to evaluate spatial reasoning. These allow simple quantitative evaluation per data point, over metrics commonly used in indoor scene synthesis that compute distributional similarities or discriminatory capability of a classification model between a generated set of rooms and a test set of rooms similar to the training dataset. In the following subsections, we describe each task and corresponding questions.

## 3.1 S1. COMMUNICATING SPATIAL COORDINATES WITH A VLM

In the first task, we focus on evaluating the VLMs' ability to communicate spatial locations effectively. This task requires the models to determine the positions of specific points in a room defined using relative relationships in text. Specifically, we ask to compute positions of corners of the room

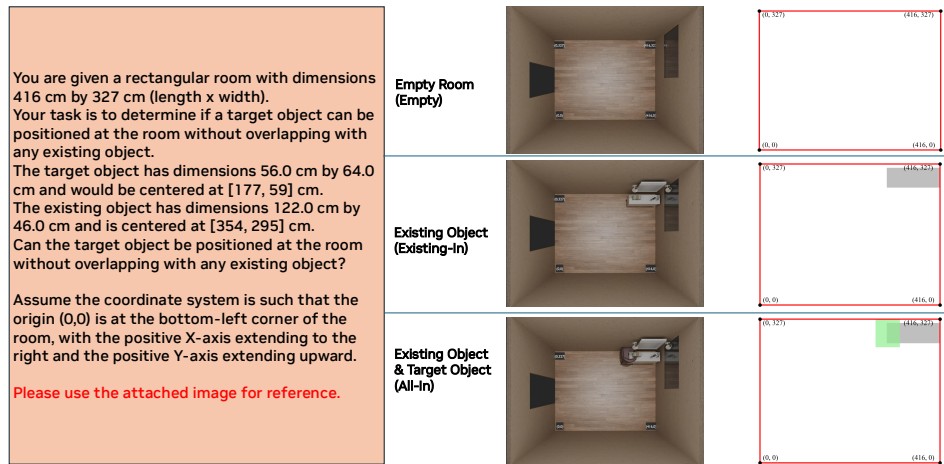

Figure 2: Example question for S2. free space reasoning with each of the three levels of complexity, showing 3D1 (top-down) rendering and sketches, with absolute coordinates

and to compute positions at an offset to the left or right to an input position. To remind the reader, we test VLMs on this task using absolute coordinates, normalized coordinates, references to discrete markers within the environment (*marks*) and using *marks+text*. All tasks are set within a standardized coordinate system pre-defined to the VLM, where the origin is at the bottom-left corner of the room, the positive X-axis extends to the right, and the positive Y-axis extends upward. By analyzing the models' performance across various visual inputs and coordinate representations, we aim to identify effective methods to accurately and efficiently communicate spatial locations with VLMs. Fig. 1 shows an example using the *marks* and the *marks+text* representation. These questions are intentionally kept simple since we find not all models perform strongly on this straightforward task. For each visual axis (five: none, sketches, 3D1, 3D2, 3D3), coordinate representation (four: absolute, normalized, marks, marks+text) and question type (three: corner, left, right) we evaluate using 25 questions for a total of 1500 questions per model.

## 3.2 S2. EVALUATING FREE-SPACE REASONING

An indoor scene design agent must possess the ability to reason about free space and avoid object collisions when placing objects within a given environment. Our task asks the model to determine whether (yes/no, binary question) a target object of a given size can be placed at a given location without colliding with other existing objects in the scene. For simplicity, we test with only one pre-existing object and one target object. We consider three levels of visual difficulty in this scenario, which also mimics scenarios and model cost restrictions in which such VLM agents might be deployed. For S2, in all cases, textual description of the location and size of existing and target objects is also provided, which can allow symbolic computation of the answer along with visual hints. The three levels are as follows: **Empty**: The model reasons about the scene using a rendering of an empty room with only textual descriptions of objects; no visual information about objects is provided. It must infer potential collisions based solely on text and visual imagination. **Existing-In**: The model is given an image showing the existing object already placed in the room, but the target object is hidden. It must infer potential collisions based on text, the visual arrangement of the existing objects and visual imagination of the target object. In a scenario where an assistant works alongside a human creator, this would be the visual available to the model to compute free locations for a new requested object. **All-In**: The given image shows both the existing object and the target object already placed. Here, the solution is visually evident, as any collisions or spatial constraint violations can be directly observed in the image. This represents a scenario where an agent might be tasked to identify issues in a scene or an expensive scenario where a scene design agent gets to render the target object at many candidate locations for visual inspection, before choosing an appropriate placement location. Fig. 3 shows all rendering views for the *All-In* case in S2.

To construct this data, we choose random rectangular bedrooms from the 3D-FRONT (Fu et al., 2021) subset used by LayoutGPT (Feng et al., 2024). We select one random object (from objects placed on the ground) from the ground truth scene as the existing object. We exhaustively compute

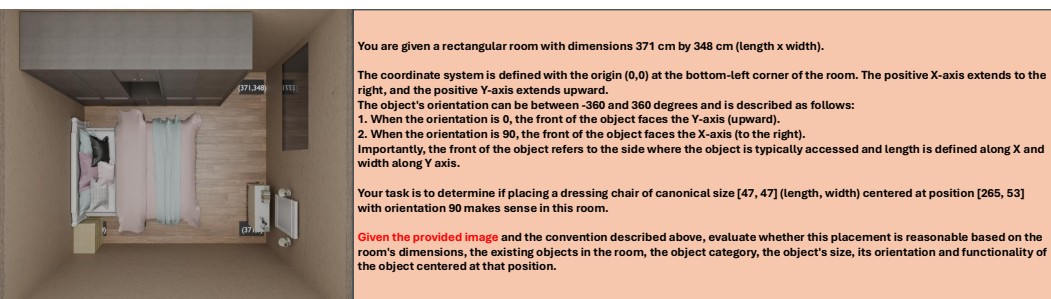

Figure 3: Visualization of the different visual inputs for S2. free space understanding (All-in). From left to right: None (Text Only), Sketches, 3D1 (top-down), 3D2 (perspective), 3D3 (embodied)

Figure 4: Example S3. joint free-space, alignment, orientation and functionality reasoning question with 3D1 (top-down) rendering and absolute coordinate representation. The model must visually reason that the requested dressing chair aligns perfectly with the existing dressing table due to its orientation and does not spatially intersect and functionally block the bed.

locations where another random object from the room would collide or not collide with the existing object and choose one position each randomly to get two questions per room, one with a collision and one without. We randomly choose one of the two questions to add to our dataset generating 25 questions per visual axis, coordinate representation and question type for a total of 1500 questions per model. Fig. 2 and Fig. 3 show examples of S2 questions.

## 3.3 S3. EVALUATING JOINT FREE-SPACE, ALIGNMENT, ORIENTATION AND FUNCTIONALITY REASONING

In our final task, we evaluate the ability of VLMs to jointly reason about free-space, object alignment, orientation, and functionality when placing a target object within a given room. Specifically, we test whether a VLM can determine whether it is appropriate (binary, yes/no) to place a target object of a given category, size, at a given location and orientation (different from S1 and S2, we also provide an orientation which can affect functionality) in a room that already contains various other objects. We view this test as one step of an iterative indoor scene layout design agent. The models are provided with textual information about the room's dimensions, a standardized coordinate system, and a list of existing objects with their center coordinate or nearest marker in the case of markers. Differently from S2, we do not provide existing object sizes in text. This makes the task primarily visual, since pure symbolic computation for free-space checking cannot be performed as object sizes must be visually inspected and approximated. In S3, we consider well populated rooms as opposed to S2.

Specifically, we again choose random rectangular bedrooms from 3D-FRONT. We keep all but one GT object on the floor in the data as the existing objects in the scene. The left-out object is used as the target object. We use its GT location and orientation in the scene as the appropriate placement example and manually annotate a new location and orientation in the same scene as an inappropriate placement. Inappropriate placements in the data can be due to functionality (orientation obstructing access to object, obstructing access to other furniture etc.) or alignment (dressing table and dressing chair not placed together as a group, bed floating in the middle of the room instead of being aligned with the wall and nightstands) or free-space collisions. Fig. 4 shows an example question from S3.

In this experiment, we test with only existing objects rendered in the scene. We skip the embodied view (3D3) since we find occlusions can make some questions unanswerable. We also skip sketches (understanding functionality requires more than rendering an object as a 2D box) and no images (no

symbolic computation possible without object sizes given in text) along the visual evaluation axes to focus only on rendered rooms in 3D1 (top-down) and 3D2 (perspective). We utilize 50 questions (25 appropriate, 25 inappropriate) along each visual axis (3D1 and 3D2) and coordinate representation for a total of 400 questions per model in this experiment.

# 4 EVALUATION AND ANALYSIS

**Models.** In this section, we discuss and analyze the performance of five representative $\{vision, text\} \rightarrow \{text\}$ VLMs on our three tasks (Sec. 3) across the proposed visual and co-ordinate representation axes. We test GPT-4o (OpenAI, 2024) (gpt-4o-2024-05-13 on the API) and GPT-4o-mini (gpt-4o-mini-2024-07-18 on the API) as the frontier proprietary models. Along with the GPT-4o models, we consider three recently released and competitive open (source / weight) VLMs: Llama-3.2 (90b-vision-instruct) (MetaAI, 2024), LLaVA-NeXT (Qwen-1.5-110B) (Li et al., 2024) and Qwen2-VL (72B-Instruct) (Wang et al., 2024b). We note that smaller models such as Cambrian1 (Tong et al., 2024a) (example in supplementary), LLaVA-NeXT (Llama-3-8B) (Li et al., 2024) performed poorly on our S1 (Sec. 3.1) task, leading to our focus on large models with strong reasoning capabilities based on public benchmarks.

Next, we analyse quantitative and qualitative results from tasks S1 (Sec. 3.1), S2 (Sec. 3.2) and S3 (Sec. 3.3) and offer insights. The prompts used are detailed in the supp. material. Since our qualitative results include showing long reasoning paths followed by models, we move them to the supp. material to save space and refer to them in the following text.

## 4.1 S1. COMMUNICATING SPATIAL COORDINATES WITH A VLM

To remind the reader, we investigate ways of communicating co-ordinates with VLMs in S1 (Fig. 1, Sec. 3.1) by addressing three straightforward questions: identifying the room's corner locations and determining left/right offsets from a given location. Table 1 shows results for S1 when averaging over all visual axes (None, Sketches, 3D1 (top-down), 3D2 (perspective) and 3D3 (embodied)). The full table of results is provided in the supp. material in Table 6.

Table 1: S1 Communicating coordinates: Accuracy of models by coordinate representation

| Model | Absolute | Marks | Marks + Text | Normalized |
|---|---|---|---|---|
| GPT-4o | 0.989 | 0.027 | 0.923 | 1.000 |
| GPT-4o-mini | 0.987 | 0.067 | 0.803 | 1.000 |
| LLaVA-NeXT-110B | 0.504 | 0.050 | 0.173 | 0.901 |
| LLaMA-3.2-90B | 0.952 | 0.340 | 0.731 | 1.000 |
| Qwen2-VL-72B | 0.456 | 0.340 | 0.581 | 0.981 |
| **Average** | 0.778 | 0.165 | 0.642 | **0.977** |

We observe that even on this simple task, there is discrepancy in performance across models and coordinate representation. Across the board, *normalized coordinates* yield the best performance with consistent full scores for models like GPT-4o , GPT-4o-mini , and LLaMA-3.2-90B , with LLaVA-NeXT-110B and Qwen2-VL-72B making few errors in computing left/right offsets. This is however untrue for absolute coordinates, where we find larger probability of error in models when performing arithmetic operations. With *marks*, we observe weak performance across S1, with models sometimes misunderstanding marks (Fig. 9) and falling back to coordinates. We see that models most often ignore the regular grid of markers in the visual input (see Figure 17), despite being prompted, and can fall back to language biases such as assuming sequential order of markers to add or subtract one for left/right marker questions (Figure 18). LLaMA-3.2-90B and Qwen2-VL-72B do best with marks, but with low overall success (34%). With *marks+text*, we test the case where perfect OCR for markers on the image are provided to the model as text and still find that models other than GPT-4o (92.3%) perform unreliably. While markers might succeed in improving object grounding (Yang et al., 2023; Liao et al., 2024a), we find no evidence that they help in referring to spatial locations in our experiment. We also show an example of Cambrian-34B failing with *marks+text* in the supp. material (Fig. 16).

With respect to the visual axis, we notice marginally small effects of change in visual representation in Table. 6. This is perhaps to be expected since all answers can be computed with text only (other

than with *marks*), as evidenced by similar performance when the models are not provided any images ("None" on the visual axis).

We note here that our results apply to a small bounded scene scenario. In large unbounded scenes, absolute or normalized coordinates might become too large or lose precision respectively. Instead of dense markers used in our case, sparse markers could be placed across an unbounded scene for the model to reference spatial coordinates. Whether our results on communication of coordinates hold for large scenes is unknown and left for future work.

### 4.2 S2. EVALUATING FREE-SPACE REASONING

In S2 (Sec. 3.2), we evaluate VLMs' ability to reason whether adding a target object into a scene with an existing object would results in a collision (Fig. 2). We once again perform our experiment across all five visual and four location representations. The experiment tests three different visual difficulty levels, *empty, existing-in and all-in* as described in Sec. 3.2. Table. 2 presents results averaged over visual and location representations, Table. 3 shows results per visual axis and Table. 8 in supplementary presents more complete results, averaged over representations. We note that since all questions are binary, 0.5 performance is equivalent to random guessing.

Table 2: S2 Free space reasoning: Accuracy of models by question type. Bold shows best performing question-type per model. Models perform better with less visual information.

| Model | All-in | Empty | Existing-in | Average |
|---|---|---|---|---|
| GPT-4o | 0.738 | **0.766** | 0.718 | 0.742 |
| GPT-4o-mini | 0.618 | **0.674** | 0.593 | 0.631 |
| LLaVA-NeXT-110B | **0.533** | 0.509 | 0.468 | 0.504 |
| LLaMA-3.2-90B | 0.585 | **0.606** | 0.525 | 0.574 |
| Qwen2-VL-72B | **0.553** | 0.518 | 0.543 | 0.536 |

Table 3: S2 Free space reasoning: Accuracy of models across different visual axes. Bold shows best performing visual axis per model. Models prefer text-only inputs and fail to reason visually.

| Model | None (text) | Sketches | 3D1 | 3D2 | 3D3 | Visual Average |
|---|---|---|---|---|---|---|
| GPT-4o | **0.853** | 0.780 | 0.697 | 0.703 | 0.760 | 0.735 |
| GPT-4o-mini | **0.707** | 0.677 | 0.603 | 0.603 | 0.620 | 0.626 |
| LLaVA-NeXT-110B | **0.613** | 0.500 | 0.527 | 0.483 | 0.477 | 0.497 |
| LLaMA-3.2-90B | **0.653** | 0.540 | 0.573 | 0.540 | 0.623 | 0.569 |
| Qwen2-VL-72B | 0.533 | **0.573** | 0.550 | 0.477 | 0.547 | 0.537 |

We immediately notice in Table. 2 that free space reasoning is challenging, with all models having greater than 20% chance on average of making errors. GPT-4o performs best over all models across all three difficulty levels, averaging 74.2% accuracy, while LLaVA-NeXT-110B and Qwen2-VL-72B perform at chance level. On analysis, **we find that all models tend to resort to symbolically compute free space, making arithmetic or logical errors in computing collisions** (Fig. 10, showing GPT-4o , the best performing model in our study). The same figure shows how in the *All-in* case, the visual input clearly shows no overlap, yet GPT-4o ignores the image and resorts to wrong symbolic computation. This is reflected in the surprising result that having no image (Table. 3) improves over every form of visual input to the model, for all models except Qwen2-VL-72B , which itself has close to chance performance on this task. GPT-4o reasons with 85.3% accuracy with no images, with 78% accuracy with simple sketches and has lower accuracy across all other visual axes with detailed rendering. Hence, we find that not only do models tend to resort to symbolic computation in the face of apparent answers in an image, the presence of an image degrades symbolic reasoning ability, while following the same reasoning path. We remind the reader that the only difference between these cases is the presence of an image and additional prompting to look at the image, the objects and the room are exactly the same per data point across visual axes in the evaluation dataset. Fig. 10 and Fig. 11 demonstrate this on the same example. Finally, with respect to the kind of rendering (3D1, 3D2, 3D3) preferred by the models, we do not observe a clear winner. In fact, **we find the GPT models consistently perform better with simplified sketches of the room as opposed to any detailed rendering, while reasoning best without any image at all**. We note here that we do

prompt the models to look at the image. Attempts to prompt more aggressively had limited success. Specifically, we tried multiple phrasings and started from prompts used by Layout-GPT (Feng et al., 2024) and Holodeck (Yang et al., 2024). We also tried zero-shot CoT prompting (Kojima et al., 2023; Wei et al., 2023) and tested with Set-of-marks visual prompting (Yang et al., 2023). Notably, we observed using CoT resulted in no performance change (Tab. 13). We hope this guides the next generation of visual instruction tuning datasets to improve visual reasoning in such contexts.

Since we find that the VLMs we study do not elicit visual reasoning and instead opt for symbolic reasoning in text when possible, we continue with two paths. First, we design our S3 task (Sec. 3.3) to be unsolvable with text-based symbolic reasoning by removing object details from the text prompt, forcing the model to grok them from the image. Second, we discuss how today's VLMs might be aided by offloading symbolic computation into **tools** in the form of pre-defined functions, which we describe next.

**Free-space reasoning with tools.** We perform a small experiment where the GPT family of models that support tool usage through their API, are provided with a pre-defined function to compute intersections as an additional tool. Table. 4 shows that this lets GPT-4o get perfect accuracy with normalized and absolute representation, while some mistakes in choosing right function parameters(Table. 9 in supplementary shows accuracy of parameters in tool calls) leads to performance degradation with marks. Note that for marks, we let the model use a tool that also accepts marks as input parameters and hence the model does not need to convert marks into coordinates, which we expect to fail from S1 and S2 results. GPT-4o-mini lags behind slightly, but still achieves a perfect score with normalized coordinates. While beyond the scope of visual reasoning that we aim to study in this paper, this experiment shows how practitioners could offload computation from LLMs into functions for spatial reasoning tasks when possible. In a production scenario, we would look to combine both visual reasoning and tool calling to offload computation for stronger spatial reasoning.

Table 4: Performance on S2 with Tool Usage

| Model | Absolute | Marks | Marks + Text | Normalized | Average |
|---|---|---|---|---|---|
| GPT-4o | **1.00** | **0.93** | **0.87** | **1.00** | **0.95** |
| GPT-4o-mini | 0.97 | 0.74 | 0.78 | 1.00 | 0.87 |

### 4.3 S3. EVALUATING JOINT FREE-SPACE, ALIGNMENT, ORIENTATION AND FUNCTIONALITY REASONING

In our final task, S3 (Sec. 3.3), we challenge VLMs with binary questions regarding the suitability of placing a piece of furniture in an existing well furnished room. Following our observations in S2 4.2 that VLMs over rely on symbolic reasoning through text even in the face of clear visual clues, we remove textual information about existing objects in the scene to force visual reasoning. We expect this task to be extremely challenging, and hence add two prompting techniques to help the model reason better. In the *Explicit Scene Graph* case, we take inspiration from Holodeck Yang et al. (2024) (that generates intermediate scene graphs using an LLM) and Least-to-most prompting Zhou et al. (2022) to prompt the model to first describe the scene with a scene graph (given an example format), followed by reasoning. To aid the model further, we provide the models with object categories and their center location in the room, referred to as *Explicit Scene Graph w/ Text Redundancy*. In this case with marks, we also provide the matrix of marks representing the room in text. The base case where the question is directly posed without additional prompting is referred to as *Implicit*. The prompts are shown in the supp. material across qualitative examples.

Table. 11 summarizes the results, where we confirm the challenging nature of this task by observing chance performance across all models, regardless of prompting. We find various compounding sources of error in the models through qualitative analysis. First, we find that while LLMs might *generate* scene graphs well Yang et al. (2024), VLMs hallucinate while *perceiving* similar scene graphs from images(Fig. 15). VLMs seem to understand orientation of objects well, but it can lead to errors in reasoning about their spatial extent(Fig. 15). Improper reasoning of locations leads to reasoning errors(Fig. 12). We also find that despite the difficulty of the task with multiple possible sources of error, GPT-4o can sometimes reason well as shown in 14, with only a slight error in the perceived scene graph. **While all models perform at chance, we find the GPTs and LLaMA-3.2-90B perform admirably at describing a room as a scene graph**. While a variety of multi-agent VLM based pipelines could be designed for this task –a very welcome contribution– we hope to

see a VLM that can perform such joint reasoning well off-the-shelf, making it an ideal candidate to build 3D design assistants with.

Table 5: S3 Joint Reasoning: Accuracy of models by prompting technique. Refer to Sec. 4.3 for a description of prompting techniques used as headings

| Model | Implicit | Explicit Scene Graph | Explicit Scene Graph w/ Text Redundancy | Average |
|-------|----------|----------------------|------------------------------------------|---------|
| GPT-4o | 0.513 | 0.397 | 0.500 | 0.457 |
| GPT-4o-mini | 0.470 | 0.480 | 0.457 | 0.469 |
| LLaVA-NeXT-110B | 0.483 | 0.480 | 0.480 | 0.481 |
| LLaMA-3.2-90B | 0.513 | 0.503 | 0.500 | 0.506 |
| Qwen2-VL-72B | 0.467 | 0.490 | 0.457 | 0.471 |

## 5 DISCUSSION AND CONCLUSION

**Open Sources Vs Proprietary Models** We find both open-source and proprietary models exhibit an over-reliance on language processing, performing better with textual inputs alone. In S1, both models achieve perfect accuracy using normalized coordinates, confirming the effectiveness of this representation. Beyond normalized coordinates, GPT-4o performs well with marks + text, while LLaMA-3.2-90B performs better with Marks alone, suggesting that GPT-4o may rely more on textual descriptions. GPT-4o outperforms LLaMA-3.2-90B across all question types in S2 in free-space checking and both perform at chance. We find Qwen2-VL-72B and LLaVA-NeXT-110B lag behind the GPTs and LLaMA-3.2-90B across our evaluation.

**Limitations.** While we intend to study 3D reasoning, our tasks could be considered to be reasoning 2D (even though we present 3D renders) since we do not consider reasoning of objects on top of each other in our data. Since our data is already challenging for state-of-the-art VLMs, we leave this additional complexity for future work. Our work also applies to an indoor closed room setting, in rectangular rooms, similar to existing work in LLM-based scene synthesis Feng et al. (2024). In open and freely shaped scenes, these results might not hold. Yet, we believe our data and findings could be useful for shaping the next generation of VLMs. Finally, we do not consider VLMs fine-tuned specifically for 3D, such as Yuan et al. (2024); Cheng et al. (2024); Chen et al. (2024), in our evaluation. These models may indeed perform better on our tasks. In the pursuit of general purpose design assistants, and since our tasks are solvable from a single image only, we stick to testing large pre-trained VLMs off-the-shelf with single image inputs. and focused our evaluation of GPT-4o and GPT-4o-mini as frontier API-access models and evaluated LLAMA-3.2-90B, LLaVA-Next-110B and Qwen-VL-72B as frontier open access models. We leave evaluating fine-tuned VLMs to future work. Constructing fine-tuned VLMs that serve as 3D-design assistants is an exciting avenue for future research.

In conclusion, we systematically evaluate VLMs on reasoning tasks using indoor scene synthesis as testbed, revealing insights into their capabilities and limitations. We found that VLMs consistently perform better when using normalized coordinates for spatial communication, as opposed to absolute coordinates or image markers, indicating that standardized representations could enhance spatial understanding. Interestingly, VLMs performed best without visual input, outperforming their results with sketches or detailed renderings (Tab. 10). This suggests an over-reliance on language and indicates that SoTA VLMs do not effectively utilize visual information for spatial reasoning, similar to findings in Tong et al. (2024b); Rahmanzadehgervi et al. (2024a). Free space reasoning is a challenge for VLMs, with models performing only marginally better than random guessing. The tendency of VLMs to compute free space mathematically—often incorrectly—even when visual cues indicate collisions underscores their limitations in visual perception. However, integrating tools like collision checking significantly improved performance, as seen with models like GPT-4o. Compounded errors in a complex reasoning task involving free space, orientation, object alignment, and functionality reasoning led to chance performance, highlighting visual reasoning limitations of today's VLMs.

We hope these findings help to illustrate both the potential and limitations of using VLMs off-the-shelf for 3D reasoning tasks and underscore the need for improved training methodologies and data curation to develop intelligent 3D design assistants that can leverage both visual and linguistic information.

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

# A    Supplementary Material

In this supplementary material, we provide additional experimental results, tables, and analysis supporting the main paper. The content is organized into the following sections:

- **Section B.1.** In this section, we explore how models handle the communication of spatial coordinates. The additional experiments expands on the ability of the model to reason about spatial positioning and accurately communicate coordinates.

- **Section B.2.** Here, we assess the models' capabilities in reasoning about free space. This involves determining whether objects can be placed in a given position without collision. We additionally evaluate the accuracy of parameters passing when using tools (e.g. collision detection), which provides a stricter evaluation of the model's tool execution capabilities.

- **Section B.3.** This section expands the main tables of the paper for S3 (e.g. more complex reasoning tasks that require models to jointly evaluate free-space availability, object alignment, orientation, and functionality).

- **Section C.** This section details the experimental setup, including the hardware used (such as GPUs), software libraries (Python, PyTorch), and the specific model checkpoints employed in the experiments. It also distinguishes between model used via API access.

- **Section D.** In this final section, we provide qualitative examples from the experiments, illustrating full prompts and the reasoning paths taken by the models. Both success and failure cases are presented to highlight the strengths and limitations of the models.

# B    Additional Experimental Results and Complete Tables

Here, we expand on the experimental results and analysis illustrated on Section 4, and show the complete experimental results.

## B.1    S1. Communicating Spatial Coordinates with a VLM

Table 7 illustrates the performance of the models across different question types, with results averaged over all visual axes and representations. Overall, we observe that "corners" type of question tends to be more challenging for the models. Notably, LLaMA-3.2-90B outperforms GPT-4o on two out of three question categories (e.g., left and right) and shows a slight advantage in the overall average performance.

Table 6, on the other hand, presents the average accuracy results across different models and representations, evaluated on S1 averaged over the three question types. The final column represents the grand total, which is the aggregate across all representation types. For each model, the highest values within a given row are highlighted in bold, emphasizing the best performance in each category. For instance, GPT-4o demonstrates a near-perfect normalized total of 1.000 across all representations. Open source models such as LLaMA-3.2-90B , Qwen2-VL-72B and LLaVA-NeXT-110B exhibit more variability in their results, with LLaMA-3.2-90B achieving higher accuracy on sketches.

## B.2    S2. Evaluating free-space reasoning

Table 8 presents the average accuracy for each model and question type in S2. The category "none" refers to text-only inputs, where no visual information is provided. Notably, the table reveals the models' strong reliance on language, particularly in the GPT-4 family. Interestingly, the models' performance does not show significant improvement even when the answers are primarily derived from visual information, suggesting that their ability to leverage visual inspection remains limited.

Table 9 presents the accuracy of parameter selection when calling the tool, assessing the model's ability to correctly pass the appropriate parameters. This provides a stricter evaluation of the model's tool execution capabilities. In our case, we observed that the model sometimes incorrectly swapped the target and source object coordinates when calling the collision detector. While the results remained correct in these instances—since the function's output did not change—this could present issues in more complex scenarios.

Table 6: S1 Accuracy by model and representation

| Model | Representation | 3D1 | 3D2 | 3D3 | none | sketches | Average |
|---|---|---|---|---|---|---|---|
| | absolute | 0.987 | 0.987 | 0.987 | **1.000** | 0.987 | 0.989 |
| | marks | **0.027** | **0.027** | **0.027** | | **0.027** | **0.027** |
| GPT-4o | marks + text | 0.907 | 0.907 | 0.907 | 0.987 | 0.907 | 0.923 |
| | normalized | **1.000** | **1.000** | **1.000** | **1.000** | **1.000** | **1.000** |
| | Average | 0.730 | 0.730 | 0.730 | **0.996** | 0.730 | 0.772 |
| | absolute | **0.987** | **0.987** | **0.987** | 0.987 | **0.987** | **0.987** |
| | marks | **0.067** | **0.067** | **0.067** | | **0.067** | **0.067** |
| GPT-4o-mini | marks + text | 0.747 | **0.827** | **0.827** | 0.813 | 0.800 | 0.803 |
| | normalized | **1.000** | **1.000** | **1.000** | **1.000** | **1.000** | **1.000** |
| | Average | 0.700 | 0.720 | 0.720 | **0.933** | 0.713 | 0.748 |
| | absolute | 0.507 | 0.533 | 0.507 | 0.387 | **0.587** | 0.504 |
| | marks | 0.027 | 0.013 | 0.040 | | **0.120** | 0.050 |
| LLaVA-NeXT-110B | marks + text | 0.173 | 0.120 | 0.160 | **0.253** | 0.160 | 0.173 |
| | normalized | **0.960** | 0.907 | 0.907 | 0.907 | 0.827 | 0.901 |
| | Average | 0.417 | 0.393 | 0.403 | **0.516** | 0.423 | 0.426 |
| | absolute | **0.960** | **0.960** | **0.960** | 0.920 | **0.960** | 0.952 |
| | marks | 0.413 | 0.080 | 0.400 | | **0.467** | 0.340 |
| LLaMA-3.2-90B | marks + text | 0.813 | 0.627 | 0.560 | 0.747 | **0.907** | 0.731 |
| | normalized | **1.000** | **1.000** | **1.000** | **1.000** | **1.000** | **1.000** |
| | Average | 0.797 | 0.667 | 0.730 | **0.889** | 0.833 | 0.778 |
| | absolute | 0.427 | 0.453 | 0.467 | **0.520** | 0.413 | 0.456 |
| | marks | 0.387 | 0.027 | 0.347 | | **0.600** | 0.340 |
| Qwen2-VL-72B | marks + text | 0.680 | 0.440 | 0.573 | 0.533 | 0.680 | 0.581 |
| | normalized | 0.973 | 0.987 | 0.987 | **1.000** | 0.960 | 0.981 |
| | Average | 0.617 | 0.477 | 0.593 | **0.684** | 0.663 | 0.603 |
| Average | | 0.652 | 0.597 | 0.635 | **0.804** | 0.673 | 0.665 |

Table 7: Average accuracy of models across different question categories for S1.

| Model | Corners | Left | Right | Average |
|---|---|---|---|---|
| GPT-4o | 0.728 | **0.806** | 0.781 | 0.772 |
| GPT-4o-mini | 0.636 | **0.806** | 0.802 | 0.748 |
| LLaVA-NeXT-110B | **0.442** | 0.417 | 0.419 | 0.426 |
| LLaMA-3.2-90B | 0.674 | **0.836** | 0.823 | **0.778** |
| Qwen2-VL-72B | **0.625** | 0.598 | 0.585 | 0.603 |
| Average | 0.621 | **0.693** | 0.682 | 0.665 |

From the table, we can see that GPT-4o demonstrates near-perfect accuracy in calling the tool with the correct parameters. In contrast, GPT-4o-mini struggles, particularly when the representation is not normalized. This aligns with our previous finding that normalized coordinates lead to better performance for the model, holding true also for function calling.

## B.3 S3. EVALUATING JOINT FREE-SPACE, ALIGNMENT, ORIENTATION AND FUNCTIONALITY REASONING

Table 12 reports the accuracy of the models across different question types and visual axis categories. We observe that the models' performance hovers around random chance for binary yes/no questions. Neither explicit instructions to construct a scene graph nor the use of different visual axes seem to

Table 8: S2. Average accuracy by model and question type

| Model | Visual Axis | All in | Empty | Existing in | Average |
|---|---|---|---|---|---|
| GPT-4o | 3D1 | 0.680 | 0.750 | 0.660 | 0.697 |
| | 3D2 | 0.730 | 0.710 | 0.670 | 0.703 |
| | 3D3 | 0.740 | 0.790 | 0.750 | 0.760 |
| | none | | **0.853** | | 0.853 |
| | sketches | 0.800 | 0.750 | 0.790 | 0.780 |
| | Average | 0.738 | 0.766 | 0.718 | 0.742 |
| GPT-4o-mini | 3D1 | 0.610 | 0.660 | 0.540 | 0.603 |
| | 3D2 | 0.590 | 0.630 | 0.590 | 0.603 |
| | 3D3 | 0.600 | 0.680 | 0.580 | 0.620 |
| | none | | **0.707** | | 0.707 |
| | sketches | 0.670 | 0.700 | 0.660 | 0.677 |
| | Average | 0.618 | 0.674 | 0.593 | 0.631 |
| LLaVA-NeXT-110B | 3D1 | 0.610 | 0.490 | 0.480 | 0.527 |
| | 3D2 | 0.540 | 0.460 | 0.450 | 0.483 |
| | 3D3 | 0.470 | 0.500 | 0.460 | 0.477 |
| | none | | **0.613** | | 0.613 |
| | sketches | 0.510 | 0.510 | 0.480 | 0.500 |
| | Average | 0.533 | 0.509 | 0.468 | 0.504 |
| LLaMA-3.2-90B | 3D1 | 0.610 | 0.620 | 0.490 | 0.573 |
| | 3D2 | 0.580 | 0.570 | 0.470 | 0.540 |
| | 3D3 | 0.620 | **0.680** | 0.570 | 0.623 |
| | none | | 0.653 | | 0.653 |
| | sketches | 0.530 | 0.520 | 0.570 | 0.540 |
| | Average | 0.585 | 0.606 | 0.525 | 0.574 |
| Qwen2-VL-72B | 3D1 | 0.570 | 0.510 | 0.570 | 0.550 |
| | 3D2 | 0.500 | 0.470 | 0.460 | 0.477 |
| | 3D3 | 0.590 | 0.520 | 0.530 | 0.547 |
| | none | | 0.533 | | 0.533 |
| | sketches | 0.550 | 0.560 | **0.610** | 0.573 |
| | Average | 0.553 | 0.518 | 0.543 | 0.536 |
| Average | | **0.605** | 0.615 | 0.569 | 0.597 |

improve the performance beyond random chance. Lower accuracy values are highlighted in red for emphasis.

We additionally provide example of the prompts used for the S3 questions in figures 5, 6, and 7.

## C  EXPERIMENTAL DETAILS AND MODEL CHECKPOINTS

For most open-source models, such as Llava and Qwen-2-VL, we utilized the pre-trained check-points available on Hugging Face. Our experiments were conducted using 8 NVIDIA A100 GPUs, with implementations based on Python 3.10 and PyTorch 2.4. For commercial models, specifically GPT-4o and GPT-4o mini, we accessed them via the OpenAI API. For Llama-3.2 Vision, we used NVIDIA NIMs. All models were tested using their default parameter configurations.

Table 9: S2. Tool usage average of parameter accuracy

| Model | Representation | 3D1 | 3D2 | 3D3 | None | Sketches | Grand Total |
|---|---|---|---|---|---|---|---|
| GPT-4o | Absolute | 1.000 | 1.000 | 1.000 | 1.000 | 1.000 | 1.000 |
| | Marks | 0.880 | .853 | 0.867 | - | 0.853 | 0.863 |
| | Marks + Text | 0.747 | 0.760 | 0.760 | 0.720 | 0.773 | 0.757 |
| | Normalized | 1.000 | 1.000 | 1.000 | 1.000 | 1.000 | 1.000 |
| | **Average** | 0.907 | 0.903 | 0.907 | 0.907 | 0.907 | 0.906 |
| GPT-4o-mini | Absolute | 0.720 | 0.653 | 0.693 | 0.920 | 0.720 | 0.714 |
| | Marks | 0.040 | 0.000 | 0.027 | - | 0.013 | 0.020 |
| | Marks + Text | 0.040 | 0.040 | 0.120 | 0.000 | 0.013 | 0.049 |
| | Normalized | 0.893 | 0.907 | 0.867 | 0.960 | 0.920 | 0.902 |
| | **Average** | 0.423 | 0.400 | 0.427 | 0.627 | 0.417 | 0.429 |
| **Average** | | 0.665 | 0.652 | 0.667 | 0.767 | 0.662 | 0.667 |

Table 10: Model accuracy on tasks S1 and S2 across different visual axes. Strong over-reliance on Language Processing

| Model | Task | None (text) | Sketches | 3D1 | 3D2 | 3D3 | Visual Average | % Diff |
|---|---|---|---|---|---|---|---|---|
| GPT-4o | S1 | **0.996** | 0.730 | 0.730 | 0.730 | 0.730 | 0.730 | -26.7% |
| | S2 | **0.853** | 0.780 | 0.697 | 0.703 | 0.760 | 0.735 | -13.9% |
| GPT-4o-mini | S1 | **0.933** | 0.713 | 0.700 | 0.720 | 0.720 | 0.713 | -23.6% |
| | S2 | **0.707** | 0.677 | 0.603 | 0.603 | 0.620 | 0.626 | -11.4% |
| LLaVA-NeXT-110B | S1 | **0.516** | 0.423 | 0.417 | 0.393 | 0.403 | 0.409 | -20.6% |
| | S2 | **0.613** | 0.500 | 0.527 | 0.483 | 0.477 | 0.497 | -19.0% |
| LLaMA-3.2-90B | S1 | **0.889** | 0.833 | 0.797 | 0.667 | 0.730 | 0.757 | -14.9% |
| | S2 | **0.653** | 0.540 | 0.573 | 0.540 | 0.623 | 0.569 | -12.9% |
| Qwen2-VL-72B | S1 | **0.684** | 0.663 | 0.617 | 0.477 | 0.593 | 0.588 | -14.2% |
| | S2 | 0.533 | **0.573** | 0.550 | 0.477 | 0.547 | 0.537 | +0.6% |

## D QUALITATIVE EXAMPLES

In Figures 8, 9, 10, 12, and 14, we present qualitative examples for each question type, focusing on the stronger model, GPT-4o . These figures illustrate the model's reasoning process for each question. It is important to note that in all cases, the final response was obtained by continuing the dialogue and asking the model to format its answer as a Python list in JSON format. For brevity, this part is not explicitly shown in the examples.

We used the following prompt: " *Please put your answer as a list of coordinates in JSON format. Start with* ```json *and end with* ``` *(e.g.,* ```json [ [10, 150], [393, 0] ] ``` *)* " (without quotation marks), where the example in the prompt is consistent with the representation used—in this case, absolute coordinates.

Table 11: S3 Joint Reasoning: Accuracy of models by representation and prompting technique. Refer to Sec. 4.3 for a description of prompting techniques

| Model | Method | Implicit | Explicit Scene Graph | Explicit Scene Graph w/ Text Redundancy | Average |
|---|---|---|---|---|---|
| GPT-4o | Absolute | 0.480 | 0.390 | 0.500 | 0.457 |
| | Marks | 0.570 | 0.370 | 0.460 | 0.467 |
| | Normalized | 0.490 | 0.430 | 0.480 | 0.467 |
| GPT-4o Average | | 0.513 | 0.397 | 0.480 | 0.463 |
| GPT-4o-mini | Absolute | 0.460 | 0.420 | 0.440 | 0.440 |
| | Marks | 0.500 | 0.520 | 0.470 | 0.497 |
| | Normalized | 0.450 | 0.500 | 0.460 | 0.470 |
| GPT-4o-mini Average | | 0.470 | 0.480 | 0.457 | 0.469 |
| LLAVA-Next | Absolute | 0.400 | 0.420 | 0.450 | 0.423 |
| | Marks | 0.540 | 0.460 | 0.510 | 0.503 |
| | Normalized | 0.510 | 0.560 | 0.480 | 0.517 |
| LLAVA-Next Average | | 0.483 | 0.480 | 0.480 | 0.481 |
| LLaMA-3.2 | Absolute | 0.530 | 0.510 | 0.510 | 0.517 |
| | Marks | 0.500 | 0.490 | 0.490 | 0.493 |
| | Normalized | 0.510 | 0.510 | 0.500 | 0.507 |
| LLaMA-3.2 Average | | 0.513 | 0.503 | 0.500 | 0.506 |
| Qwen2-VL-72B | Absolute | 0.390 | 0.460 | 0.430 | 0.427 |
| | Marks | 0.500 | 0.500 | 0.440 | 0.480 |
| | Normalized | 0.510 | 0.510 | 0.500 | 0.507 |
| Qwen2-VL-72B Average | | 0.467 | 0.490 | 0.457 | 0.471 |

Table 12: S3 Average accuracy across different models and visual axes

| Model | Question — Prompt | 3D1 | 3D2 | Average |
|---|---|---|---|---|
| GPT-4o | Implicit Prompt | **0.533** | 0.493 | 0.513 |
| | Explicit Scene Graph Prompt | **0.413** | 0.380 | 0.397 |
| | Text Redundancy + Explicit Scene Graph Prompt | 0.467 | 0.493 | **0.480** |
| GPT-4o Average | | **0.471** | 0.456 | 0.463 |
| GPT-4o-mini | Implicit Prompt | **0.500** | 0.440 | 0.470 |
| | Explicit Scene Graph Prompt | 0.453 | **0.507** | 0.480 |
| | Text Redundancy + Explicit Scene Graph Prompt | 0.440 | **0.473** | 0.457 |
| GPT-4o-mini Average | | 0.464 | **0.473** | 0.469 |
| LLaVA-NeXT-110B | Implicit Prompt | 0.480 | **0.487** | 0.483 |
| | Explicit Scene Graph Prompt | **0.493** | 0.467 | 0.480 |
| | Text Redundancy + Explicit Scene Graph Prompt | **0.500** | 0.460 | 0.480 |
| LLaVA-NeXT-110B Average | | **0.491** | 0.471 | 0.481 |
| LLaMA-3.2-90B | Implicit Prompt | **0.527** | 0.500 | 0.513 |
| | Explicit Scene Graph Prompt | **0.513** | 0.493 | 0.503 |
| | Text Redundancy + Explicit Scene Graph Prompt | 0.493 | **0.507** | 0.500 |
| LLaMA-3.2-90B Average | | **0.511** | 0.500 | 0.506 |
| Qwen2-VL-72B | Implicit Prompt | **0.507** | 0.427 | 0.467 |
| | Explicit Scene Graph Prompt | 0.487 | **0.493** | 0.490 |
| | Text Redundancy + Explicit Scene Graph Prompt | 0.447 | **0.467** | 0.457 |
| Qwen2-VL-72B Average | | **0.480** | 0.462 | 0.471 |
| Average | | **0.484** | 0.472 | 0.478 |

Table 13: Average accuracy for GPT-4o across question types for S2 w/ and w/o zero-shot CoT (Kojima et al., 2023)

| Model | all_in | empty | existing_in | Average |
|---|---|---|---|---|
| GPT-4o | 0.738 | 0.766 | 0.718 | 0.742 |
| GPT-4o w/ Zero-Shot-CoT | 0.688 | 0.724 | 0.713 | 0.709 |

```
System: You are a helpful visual AI agent. The user will ask you questions, and you will provide the
↪    answers.

Prompt:  You are given a rectangular room with dimensions 334 cm by 436 cm (length x width).
The provided image represents the top-down view of room.

The coordinate system is defined with the origin (0,0) at the bottom-left corner of the room. The
↪    positive X-axis extends to the right, and the positive Y-axis extends upward.
The object's orientation can be between -360 and 360 degrees and is described as follows:
1. When the orientation is 0, the front of the object faces the Y-axis (upward).
2. When the orientation is 90, the front of the object faces the X-axis (to the right).

Importantly, the front of the object refers to the side where the object is typically accessed and
↪    length is defined along X and width along Y axis.

Your task is to determine if placing a nightstand of canonical size [48, 48] (length, width) centered
↪    at position [310, 372] with orientation -90 makes sense in this room.
Given the provided image and the convention described above, evaluate whether this placement is
↪    reasonable based on the room's dimensions, the existing objects in the room, the object category,
↪    the object's size, its orientation and functionality of the object centered at that position.
```

Figure 5: Example of the implicit prompt used to evaluate the model on S3. The model is expected to implicitly determine the best approach to solve the task.

```
System: You are a helpful visual AI agent. The user will ask you questions, and you will provide the
↪  answers.

User:  You are given a rectangular room with dimensions 276 cm by 413 cm (length x width).
The provided image represents the top-down view of room.

The coordinate system is defined with the origin (0,0) at the bottom-left corner of the room. The
↪  positive X-axis extends to the right, and the positive Y-axis extends upward.
The object's orientation can be between -360 and 360 degrees and is described as follows:
1. When the orientation is 0, the front of the object faces the Y-axis (upward).
2. When the orientation is 90, the front of the object faces the X-axis (to the right).

Importantly, the front of the object refers to the side where the object is typically accessed and
↪  length is defined along X and width along Y axis.

Your task is to determine if placing a double_bed of canonical size [272, 209] (length, width)
↪  centered at position [177, 177] with orientation -90 makes sense in this room.
Given the provided image and the convention described above, evaluate whether this placement is
↪  reasonable based on the room's dimensions, the existing objects in the room, the object category,
↪  the object's size, its orientation and functionality of the object centered at that position.

You will break down the task into three steps:
1. Describe the providing image and existing objects in the room. Start by visually analyzing the
↪  image and identifying the key objects and their locations in relation to each other.

<Example>
The image shows a top-down view of bedroom.
A soft gray sofa is positioned at the edge, anchoring the seating area.
In front of it, a wooden coffee table sits centrally in the middle, aligned and facing the sofa, at a
↪  near distance.
A sleek TV stand is placed at the opposite edge, far from the coffee table, and also aligned to face
↪  it.
Additionally, a modern desk is located at the edge, far from the TV stand, creating a clear
↪  separation between the workspace and relaxation area.
</Example>

2. Create a text-based scene graph of the room. Organize the room and objects into a structured
↪  format, capturing their spatial relationships.

<Example>
sofa-0 | edge
coffee table-0 | middle | near, sofa-0 | in front of, sofa-0 | center aligned, sofa-0 | face to,
↪  sofa-0
tv stand-0 | edge | far, coffee table-0 | in front of, coffee table-0 | center aligned, coffee
↪  table-0 | face to, coffee table-0
desk-0 | edge | far, tv stand-0
</Example>

3. Evaluate whether the object placement makes sense in the room (yes/no) and provide a reason for
↪  your answer.
Evaluate whether this placement is reasonable based on the text-based scene graph of the room, the
↪  room's dimensions, the object category, the object's size, its orientation and functionality of
↪  the object centered at that position
```

Figure 6: Example of the explicit scene graph prompt used to evaluate the model on S3. Inspired by least-to-most prompting and text-based scene graph construction from Holodeck Yang et al. (2024), the model is instructed to first create a description of the scene, followed by constructing a text-based scene graph as intermediate steps.

```
System: You are a helpful visual AI agent. The user will ask you questions, and you will provide the
↪ answers.

Prompt:  You are given a rectangular room with dimensions 334 cm by 436 cm (length x width).
The provided image represents the top-down view of room.

The coordinate system is defined with the origin (0,0) at the bottom-left corner of the room. The
↪ positive X-axis extends to the right, and the positive Y-axis extends upward.
The object's orientation can be between -360 and 360 degrees and is described as follows:
1. When the orientation is 0, the front of the object faces the Y-axis (upward).
2. When the orientation is 90, the front of the object faces the X-axis (to the right).

Importantly, the front of the object refers to the side where the object is typically accessed and
↪ length is defined along X and width along Y axis.

Objects in the room, their exterior polygon coordinates as a [x,y] list.

double_bed. Polygon: [[97.5, 129.0], [97.5, 339.0], [330.5, 339.0], [330.5, 129.0]]
nightstand. Polygon: [[278.0, 81.0], [278.0, 129.0], [326.0, 129.0], [326.0, 81.0]]
wardrobe. Polygon: [[32.0, 0.0], [32.0, 66.0], [328.0, 66.0], [328.0, 0.0]]

Your task is to determine if placing a nightstand of canonical size [48, 48] (length, width) centered
↪ at position [310, 372] with orientation -90 makes sense in this room.
Given the provided image and the convention described above, evaluate whether this placement is
↪ reasonable based on the room's dimensions, the existing objects in the room, the object category,
↪ the object's size, its orientation and functionality of the object centered at that position.

You will break down the task into three steps:
1. Describe the providing image and existing objects in the room. Start by visually analyzing the
↪ image and identifying the key objects and their locations in relation to each other.

<Example>
The image shows a top-down view of bedroom. A soft gray sofa is positioned at the edge, anchoring the
↪ seating area. In front of it, a wooden coffee table sits centrally in the middle, aligned and
↪ facing the sofa, at a near distance. A sleek TV stand is placed at the opposite edge, far from
↪ the coffee table, and also aligned to face it. Additionally, a modern desk is located at the
↪ edge, far from the TV stand, creating a clear separation between the workspace and relaxation
↪ area.
</Example>

2. Create a text-based scene graph of the room. Organize the room and objects into a structured
↪ format, capturing their spatial relationships.

<Example>
sofa-0 | edge
coffee table-0 | middle | near, sofa-0 | in front of, sofa-0 | center aligned, sofa-0 | face to,
↪ sofa-0
tv stand-0 | edge | far, coffee table-0 | in front of, coffee table-0 | center aligned, coffee
↪ table-0 | face to, coffee table-0
desk-0 | edge | far, tv stand-0
<Example>

3. Evaluate whether the object placement makes sense in the room (yes/no) and provide a reason for
↪ your answer.
Evaluate whether this placement is reasonable based on the text-based scene graph of the room, the
↪ room's dimensions, the object category, the object's size, its orientation and functionality of
↪ the object centered at that position.
```

Figure 7: Example of the text redundancy + explicit scene graph prompt used to evaluate the model on S3. We begin by describing the existing objects in the room. Then, inspired by least-to-most prompting and text-based scene graph construction from Holodeck Yang et al. (2024), the model is instructed to first provide a description of the scene, followed by constructing a text-based scene graph as intermediate steps.

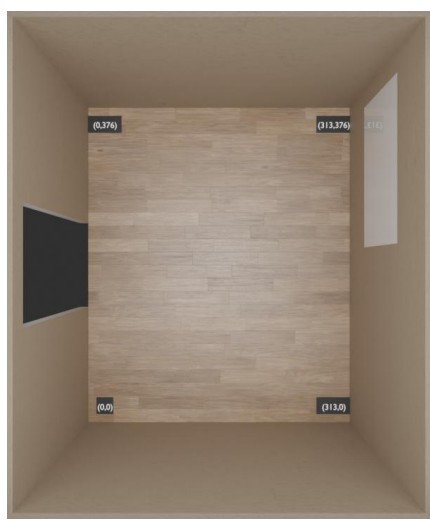

```
System: You are a helpful AI agent. You are helping a user to understand spatial relationships in a
↪  room. The user will ask you questions, and you will provide the answers.

User: You are given a rectangular room with dimensions 313 cm by 376 cm.

Your task is to determine the coordinates of the four corners in the following order: bottom-left,
↪  top-left, top-right, and bottom-right.

Assume the coordinate system is such that the origin (0,0) is at the bottom-left corner of the room,
↪  with the positive X-axis extending to the right and the positive Y-axis extending upward.

Please use the attached image for reference.

VLM Response : Given the dimensions of the rectangular room (313 cm by 376 cm) and the coordinate
↪  system where the origin (0,0) is at the bottom-left corner, the coordinates of the four corners
↪  of the room can be determined as follows:

1. **Bottom-left corner**: This is the origin of the coordinate system.
- Coordinates: (0, 0)

2. **Top-left corner**: This is directly above the bottom-left corner, at the height of the room.
- Coordinates: (0, 376)

3. **Top-right corner**: This is at the top of the room, at the width of the room.
- Coordinates: (313, 376)

4. **Bottom-right corner**: This is directly to the right of the bottom-left corner, at the width of
↪  the room.
- Coordinates: (313, 0)

So, the coordinates of the four corners in the specified order are:
- Bottom-left: (0, 0)
- Top-left: (0, 376)
- Top-right: (313, 376)
- Bottom-right: (313, 0)
```

Figure 8: Qualitative example illustrating the task for S1. Success case with corners question using the GPT-4o model.

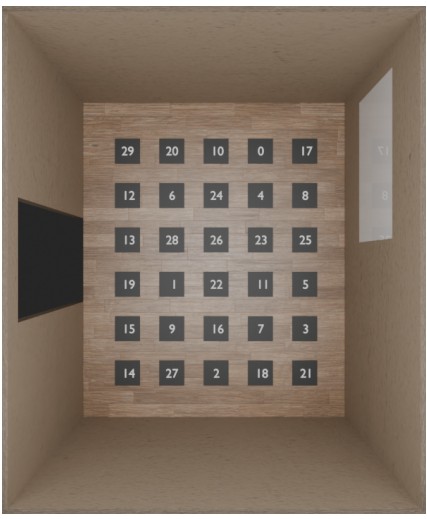

```
System: You are a helpful AI agent. You are helping a user to understand spatial relationships in a
↪   room. The user will ask you questions, and you will provide the answers.

User: You are given a rectangular room with dimensions 313 cm by 376 cm.
The provided image represents the top-down view of the rectangular room with markers.
Each marker is associated with a coordinate (x, y) within the room. The markers are placed at a
↪   distance of 53 cm from each other.

Your task is to determine the marks representing the four corners in the following order:
↪   bottom-left, top-left, top-right, and bottom-right.

Please use the attached image for reference.

VLM Response : Given the dimensions of the rectangular room (313 cm by 376 cm) and the coordinate
↪   system where the origin (0,0) is at the bottom-left corner, the coordinates of the four corners
↪   of the room can be determined as follows:

1. **Bottom-left corner**: This is the origin of the coordinate system.
- Coordinates: (0, 0)

2. **Top-left corner**: This is directly above the bottom-left corner, at the height of the room.
- Coordinates: (0, 376)

3. **Top-right corner**: This is at the top of the room, at the width of the room.
- Coordinates: (313, 376)

4. **Bottom-right corner**: This is directly to the right of the bottom-left corner, at the width of
↪   the room.
- Coordinates: (313, 0)

So, the coordinates of the four corners in the specified order are:
- Bottom-left: (0, 0)
- Top-left: (0, 376)
- Top-right: (313, 376)
- Bottom-right: (313, 0)
```

Figure 9: Illustration of a failure case for task S1. The model GPT-4o fails to correctly answer corner questions involving marks, providing the absolute coordinates of the room instead of the required corner markers.

```
System: You are a helpful AI agent. You are helping a user to understand spatial relationships in a
↪   room. The user will ask you questions, and you will provide the answers.

User: You are given a rectangular room with normalized dimensions 1 unit(s) by 1 unit(s) (length x
↪   width).
Your task is to determine if a target object can be positioned at the room without overlapping with
↪   any existing object.
The target object has dimensions 0.188 unit(s) by 0.165 unit(s) and would be centered at [0.561,
↪   0.808] unit(s).
The existing object has dimensions 0.152 unit(s) by 0.56 unit(s) and is centered at [0.281, 0.485]
↪   unit(s).

Can the target object be positioned at the room without overlapping with any existing object?

Assume the coordinate system is such that the origin (0,0) is at the bottom-left corner of the room,
↪   with the positive X-axis extending to the right and the positive Y-axis extending upward.

Please use the attached image for reference. Both existing and target objects are shown in the image

VLM Response : .... First, let's calculate the bounding boxes for both objects.
### Target Object
- Center: [0.561, 0.808]
- Dimensions: 0.188 × 0.165
The half-dimensions are:
- Half-width: 0.188/2 = 0.094
- Half-height: 0.165/2 = 0.0825
Bounding box coordinates:
- Left: 0.561 − 0.094 = 0.467
- Right: 0.561 + 0.094 = 0.655
- Bottom: 0.808 − 0.0825 = 0.7255
- Top: 0.808 + 0.0825 = 0.8905
### Existing Object
- Center: [0.281, 0.485]
- Dimensions: 0.152 × 0.56
The half-dimensions are:
- Half-width: 0.152/2 = 0.076
- Half-height: 0.56/2 = 0.28
Bounding box coordinates:
- Left: 0.281 − 0.076 = 0.205
- Right: 0.281 + 0.076 = 0.357
- Bottom: 0.485 − 0.28 = 0.205
- Top: 0.485 + 0.28 = 0.765
### Check for Overlap
To check for overlap, we need to see if the bounding boxes intersect. The conditions for no overlap
↪   are:
- The right side of one box is to the left of the left side of the other box.
- The top side of one box is below the bottom side of the other box.
Let's check these conditions:
1. **Horizontal Overlap**:
- The right side of the target object (0.655) is greater than the left side of the existing object
↪   (0.205).
- The left side of the target object (0.467) is less than the right side of the existing object
↪   (0.357).
2. **Vertical Overlap**:
- The top side of the target object (0.8905) is greater than the bottom side of the existing object
↪   (0.205).
- The bottom side of the target object (0.7255) is less than the top side of the existing object
↪   (0.765).
Since both horizontal and vertical overlap conditions are met, the target object will overlap with
↪   the existing object.
### Conclusion
The target object **cannot** be positioned at [0.561, 0.808] without overlapping with the existing
↪   object.

...

Answer: No
```

Figure 10: Illustration of a failure case for task S2 (All-IN scenario) with the GPT-4o model. The model fails to provide a correct answer, despite the fact that the problem could be easily solved through visual inspection. Instead, the model attempts a mathematical approach and fails.

```
System: You are a helpful AI agent. You are helping a user to understand spatial relationships in a
↪  room. The user will ask you questions, and you will provide the answers.

User: You are given a rectangular room with normalized dimensions 1 unit(s) by 1 unit(s) (length x
↪  width).
Your task is to determine if a target object can be positioned at the room without overlapping with
↪  any existing object.
The target object has dimensions 0.188 unit(s) by 0.165 unit(s) and would be centered at [0.561,
↪  0.808] unit(s).
The existing object has dimensions 0.152 unit(s) by 0.56 unit(s) and is centered at [0.281, 0.485]
↪  unit(s).

Can the target object be positioned at the room without overlapping with any existing object?

Assume the coordinate system is such that the origin (0,0) is at the bottom-left corner of the room,
↪  with the positive X-axis extending to the right and the positive Y-axis extending upward.

VLM Response : .... First, let's calculate the bounding boxes for both objects.

### Target Object
- Center: [0.561, 0.808]
- Dimensions: 0.188 × 0.165
The half-dimensions are:
- Half-width: 0.188/2 = 0.094
- Half-height: 0.165/2 = 0.0825
Bounding box coordinates:
- Left: 0.561 − 0.094 = 0.467
- Right: 0.561 + 0.094 = 0.655
- Bottom: 0.808 − 0.0825 = 0.7255
- Top: 0.808 + 0.0825 = 0.8905
### Existing Object
- Center: [0.281, 0.485]
- Dimensions: 0.152 × 0.56
The half-dimensions are:
- Half-width: 0.152/2 = 0.076
- Half-height: 0.56/2 = 0.28
Bounding box coordinates:
- Left: 0.281 − 0.076 = 0.205
- Right: 0.281 + 0.076 = 0.357
- Bottom: 0.485 − 0.28 = 0.205
- Top: 0.485 + 0.28 = 0.765
### Checking for Overlap
To check for overlap, we need to see if the bounding boxes intersect. The conditions for no overlap
↪  are:
- One rectangle is to the left of the other.
- One rectangle is above the other.
For no overlap:
- The right side of the target object must be less than the left side of the existing object, or
- The left side of the target object must be greater than the right side of the existing object, or
- The top side of the target object must be less than the bottom side of the existing object, or
- The bottom side of the target object must be greater than the top side of the existing object.
Let's check these conditions:
- Right side of target: 0.655
- Left side of existing: 0.205
- Left side of target: 0.467
- Right side of existing: 0.357
- Top side of target: 0.8905
- Bottom side of existing: 0.205
- Bottom side of target: 0.7255
- Top side of existing: 0.765
Checking the conditions:
- 0.655 < 0.205 (False)
- 0.467 > 0.357 (True)
- 0.8905 < 0.205 (False)
- 0.7255 > 0.765 (False)
Since one of the conditions (left side of target > right side of existing) is true, the target object
↪  can be positioned in the room without overlapping with the existing object.

### Conclusion
Yes, the target object can be positioned at [0.561, 0.808] without overlapping with the existing object
↪  centered at [0.281, 0.485].
...

Answer: Yes
```

Figure 11: Illustration of a successful case for task S2 with the GPT-4o model on the same data as in Fig. 10. When provided no images, the model follows a similar reasoning path with more accurate collision checking, resulting in a correct answer

```
User: You are given a rectangular room with dimensions 334 cm by 436 cm (length x width).
The provided image represents the top-down view of room.

The coordinate system is defined with the origin (0,0) at the bottom-left corner of the room. The
↪   positive X-axis extends to the right, and the positive Y-axis extends upward.
The object's orientation can be between -360 and 360 degrees and is described as follows:
1. When the orientation is 0, the front of the object faces the Y-axis (upward).
2. When the orientation is 90, the front of the object faces the X-axis (to the right).

Importantly, the front of the object refers to the side where the object is typically accessed and
↪   length is defined along X and width along Y axis.

Your task is to determine if placing a nightstand of canonical size [48, 48] (length, width) centered
↪   at position [310, 372] with orientation -90 makes sense in this room.
Given the provided image and the convention described above, evaluate whether this placement is
↪   reasonable based on the room's dimensions, the existing objects in the room, the object category,
↪   the object's size, its orientation and functionality of the object centered at that position.

VLM Response : To determine if placing a nightstand of size [48 cm, 48 cm] centered at position [310
↪   cm, 372 cm] with an orientation of -90 degrees makes sense in this room, we need to consider the
↪   following:

1. **Room Dimensions**: The room is 334 cm by 436 cm.
2. **Nightstand Dimensions**: The nightstand is 48 cm by 48 cm.
3. **Nightstand Position**: The nightstand is centered at [310 cm, 372 cm].
4. **Nightstand Orientation**: The front of the nightstand faces the negative X-axis (to the left)
↪   when oriented at -90 degrees.

### Step-by-Step Analysis:

1. **Position and Centering**:
- The nightstand is centered at [310 cm, 372 cm].
- This means the nightstand extends 24 cm (half of 48 cm) in all directions from its center.

2. **Boundary Check**:
- The nightstand's boundaries will be:
- Left: 310 cm - 24 cm = 286 cm
- Right: 310 cm + 24 cm = 334 cm
- Bottom: 372 cm - 24 cm = 348 cm
- Top: 372 cm + 24 cm = 396 cm

3. **Room Boundaries**:
- The room's dimensions are 334 cm by 436 cm.
- The nightstand's right boundary (334 cm) exactly touches the room's right wall.
- The nightstand's top boundary (396 cm) exceeds the room's top boundary (436 cm).

4. **Existing Objects**:
- The image shows a bed and other furniture in the room.
- The proposed position of the nightstand is very close to the bed, which might interfere with the
↪   bed's functionality and accessibility.

### Conclusion:
- **Boundary Violation**: The nightstand's top boundary (396 cm) exceeds the room's top boundary (436
↪   cm), which means it will not fit within the room's dimensions.
- **Interference with Existing Objects**: The nightstand's position is very close to the bed, which
↪   could interfere with the bed's accessibility and functionality.

Given these points, placing the nightstand at the specified position and orientation does not make
↪   sense in this room. It would be better to find a different position that does not violate the
↪   room's boundaries and does not interfere with existing furniture.

....

Answer: No
```

Figure 12: Illustration of a failure case for task S3 with the GPT-4o model.

```
User: You are given a rectangular room with dimensions 334 cm by 436 cm.
The provided image represents the top-down view of room.
Each marker is associated with a coordinate (x, y) within the room.
The markers are placed at a distance of 62 cm from each other.

The coordinate system is defined with the origin (0,0) at the bottom-left corner of the room. The
↪   positive X-axis extends to the right, and the positive Y-axis extends upward.
The object's orientation can be between -360 and 360 degrees and is described as follows:
1. When the orientation is 0, the front of the object faces the Y-axis (upward).
2. When the orientation is 90, the front of the object faces the X-axis (to the right).

Importantly, the front of the object refers to the side where the object is typically accessed and
↪   length is defined along X and width along Y axis.

Your task is to determine if placing a nightstand of canonical size [48, 48] (length, width) centered
↪   at marker '1' with orientation -90 makes sense in this room.

Given the provided image and the convention described above, evaluate whether this placement is
↪   reasonable based on the room's dimensions, the existing objects in the room, the object category,
↪   the object's size, its orientation and functionality of the object centered at that position.

VLM Response : To determine if placing a nightstand of size [48 cm, 48 cm] centered at marker '1'
↪   with an orientation of -90 degrees makes sense, we need to consider the following:

1. **Marker Coordinates**: Marker '1' is located at the bottom-left corner of the room. Given the
↪   markers are placed 62 cm apart, the coordinates of marker '1' are (62 cm, 62 cm).

2. **Nightstand Dimensions and Orientation**:
- The nightstand has dimensions of 48 cm by 48 cm.
- With an orientation of -90 degrees, the front of the nightstand faces the negative X-axis (to the
↪   left).

3. **Placement and Room Boundaries**:
- Centering the nightstand at marker '1' means its center is at (62 cm, 62 cm).
- The nightstand extends 24 cm in all directions from its center (since 48 cm / 2 = 24 cm).

4. **Boundary Check**:
- The left edge of the nightstand will be at 62 cm - 24 cm = 38 cm.
- The right edge of the nightstand will be at 62 cm + 24 cm = 86 cm.
- The bottom edge of the nightstand will be at 62 cm - 24 cm = 38 cm.
- The top edge of the nightstand will be at 62 cm + 24 cm = 86 cm.

5. **Existing Objects**:
- The image shows a bed and other furniture in the room.
- Marker '1' is near the bottom-left corner, and there appears to be a bed nearby.

6. **Functionality and Accessibility**:
- A nightstand is typically placed next to a bed for accessibility.
- Placing the nightstand at marker '1' would make it accessible from the bed.

Given these considerations, placing the nightstand at marker '1' with the specified orientation is
↪   reasonable. The nightstand will fit within the room's boundaries and will be accessible from the
↪   bed, fulfilling its intended functionality.

....

Answer: Yes
```

Figure 13: Illustration of an incorrect case for task S3, where the model's reasoning path is completely off. However, GPT-4o hallucinates and still guesses the correct binary "yes" answer. While the model's accuracy would be considered acceptable in this case, it repeatedly hallucinates and deviates from the correct reasoning process.

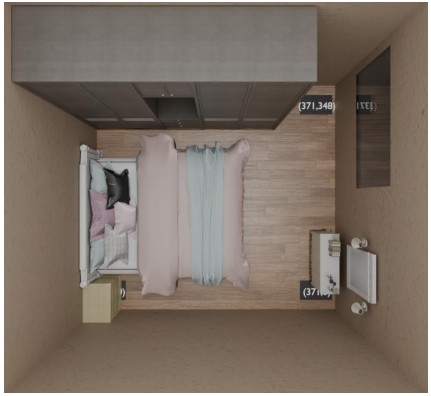

```
User: ... {See above for complete prompt examples}...
....

Your task is to determine if placing a dressing_chair of canonical size [47, 47] (length, width)
↪   centered at position [265, 53] with orientation 90 makes sense in this room.
Given the provided image and the convention described above, evaluate whether this placement is
↪   reasonable based on the room's dimensions, the existing objects in the room, the object category,
↪   the object's size, its orientation and functionality of the object centered at that position.

You will break down the task into three steps:
... {See above for complete prompt}

VLM Response : ### Step 1: Describe the provided image and existing objects in the room

The image shows a top-down view of a bedroom.
- A large bed is positioned centrally in the room, with its headboard against the top wall.
- To the left of the bed, there is a nightstand.
- At the top wall, above the bed, there is a wardrobe spanning the width of the room.
- To the right of the bed, there is a dressing table with a mirror above it.
- The floor appears to be wooden, and the walls are a neutral color.
### Step 2: Create a text-based scene graph of the room
```
bed-0 | center | headboard against, top wall
nightstand-0 | left, bed-0
wardrobe-0 | top wall | above, bed-0 | spanning width
dressing table-0 | right, bed-0 | below, mirror-0
mirror-0 | above, dressing table-0
```
### Step 3: Evaluate whether the object placement makes sense in the room
**Object to be placed: Dressing chair**
- Size: 47 cm (length) x 47 cm (width)
- Centered at position: [265, 53]
- Orientation: 90 degrees (front facing the X-axis, to the right)
**Evaluation:**
- The room dimensions are 371 cm (length) x 348 cm (width).
- The dressing chair will be centered at [265, 53], meaning its edges will extend from [241.5, 29.5]
↪   to [288.5, 76.5].
- The dressing table is located to the right of the bed, and the chair is intended to be placed in
↪   front of it.
**Reasoning:**
- The placement of the dressing chair at [265, 53] with an orientation of 90 degrees means it will
↪   face the dressing table, which is logical for functionality.
- The chair's position does not interfere with the bed or the nightstand.
- The chair's dimensions fit within the room's dimensions and do not obstruct any pathways or other
↪   furniture.
**Conclusion:**
Yes, the placement of the dressing chair centered at position [265, 53] with orientation 90 makes
↪   sense in this room. The chair is appropriately positioned in front of the dressing table,
↪   allowing for functional use without obstructing other furniture or pathways.

....

Answer: Yes
```

Figure 14: Illustration of a correct case for task S3, in the explicit scene graph case, where the GPT-4o is able to reason well about alignment, orientation and functionality, while only slightly hallucinating in the scene graph about the bed's headboard position

```
User:... {See above for complete prompt examples}...
....
Your task is to determine if placing a chair of canonical size [0.135, 0.091] (length, width)
↪   centered at position [0.648, 0.283] with orientation 65 makes sense in this room.
Given the provided image and the convention described above, evaluate whether this placement is
↪   reasonable based on the room's dimensions, the existing objects in the room, the object category,
↪   the object's size, its orientation and functionality of the object centered at that position.

You will break down the task into three steps:
1. Describe the providing image and existing objects in the room. Start by visually analyzing the
↪   image and identifying the key objects and their locations in relation to each other.

<Example>
The image shows a top-down view of bedroom. A soft gray sofa is positioned at the edge, anchoring the
↪   seating area. In front of it, a wooden coffee table sits centrally in the middle, aligned and
↪   facing the sofa, at a near distance. A sleek TV stand is placed at the opposite edge, far from
↪   the coffee table, and also aligned to face it.
Additionally, a modern desk is located at the edge, far from the TV stand, creating a clear
↪   separation between the workspace and relaxation area.
</Example>

2. Create a text-based scene graph of the room. Organize the room and objects into a structured
↪   format, capturing their spatial relationships.
<Example>
sofa-0 | edge
coffee table-0 | middle | near, sofa-0 | in front of, sofa-0 | center aligned, sofa-0 | face to,
↪   sofa-0
tv stand-0 | edge | far, coffee table-0 | in front of, coffee table-0 | center aligned, coffee
↪   table-0 | face to, coffee table-0
desk-0 | edge | far, tv stand-0
</Example>

3. Evaluate whether the object placement makes sense in the room (yes/no) and provide a reason for
↪   your answer.
Evaluate whether this placement is reasonable based on the text-based scene graph of the room, the
↪   room's dimensions, the object category, the object's size, its orientation and functionality of
↪   the object centered at that position.

VLM Response : **Step 1: Describe the providing image and existing objects in the room.**

The image shows a top-down view of a bedroom. A bed is positioned at the center, with a nightstand on
↪   each side. The bed is aligned with the longer side of the room, leaving space on either end. A
↪   dresser is placed at the edge, far from the bed, creating a clear separation between the sleeping
↪   area and the storage space. Additionally, a closet is located at the edge, far from the dresser,
↪   providing additional storage.

**Step 2: Create a text-based scene graph of the room.**

bed-0 | center
nightstand-0 | near, bed-0 | left of, bed-0
nightstand-1 | near, bed-0 | right of, bed-0
dresser-0 | edge | far, bed-0
closet-0 | edge | far, dresser-0

**Step 3: Evaluate whether the object placement makes sense in the room (yes/no) and provide a reason
↪   for your answer.**

No, the placement of the chair does not make sense in this room. The chair is centered at position
↪   [0.648, 0.283] with an orientation of 65 degrees, which means it is facing diagonally towards the
↪   top-right corner of the room. However, this placement would result in the chair being partially
↪   outside the room, as the room's dimensions are 1 unit(s) by 1 unit(s) and the chair's position is
↪   beyond the room's boundaries. Additionally, the chair's orientation does not align with any of
↪   the existing objects in the room, making it difficult to determine its intended purpose or
↪   functionality in this context.

....

Answer: NO
```

Figure 15: Illustration of an incorrect case for task S3 using the explicit scene graph prompt. Model: LLaMA-3.2-90B . Visual Axis: 3D2

```
    User: You are given a rectangular room with dimensions 366 cm by 357 cm.
    The provided image represents the top-down view of the rectangular room with markers.
    Each marker is associated with a coordinate (x, y) within the room. The markers are placed at a
    ↪   distance of 52 cm from each other.

    Determine the mark that is exactly to the left of 14.
    Ensure that the there exists indeed mark on. If no, please use [-1] to indicate it.

    You are given a rectangular room with dimensions 366 cm by 357 cm.
    The provided image represents the top-down view of the rectangular room with markers.
    Each marker is associated with a coordinate (x, y) within the room. The markers are placed at a
    ↪   distance of 52 cm from each other.
    ---
    Text Matrix:
    [['7' '17' '25' '29' '30' '8']
    ['26' '22' '4' '3' '14' '31']
    ['13' '11' '16' '2' '10' '6']
    ['34' '24' '20' '12' '5' '0']
    ['27' '33' '9' '23' '1' '18']
    ['32' '28' '19' '21' '35' '15']]]

    ---

    VLM Response : To determine the marker that is exactly to the left of 14, we need to identify the
    ↪   position of marker 14 in the matrix. Marker 14 is located in the second row and the fifth column.
    ↪   To find the marker to the left of 14, we move one column to the left, which corresponds to the
    ↪   fourth column. Looking at the second row, the marker in the fourth column is 16. Therefore, the
    ↪   marker that is exactly to the left of 14 is 16.

    ....

    Answer: ["X"]
```

Figure 16: Illustration of a failure case for task S1 (Left, Sketches) with the Cambrian-34b model using marks + text representation. We experimented with models with a smaller number of parameters, but they failed to provide the correct answer and to format it in the required JSON structure. Consequently, we focused our study on larger models with stronger reasoning capabilities, as demonstrated by public benchmarks.

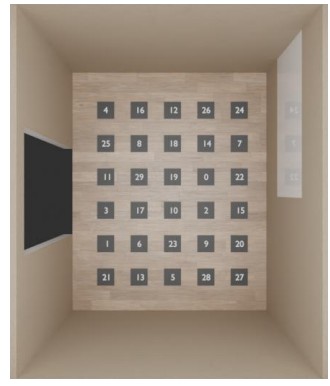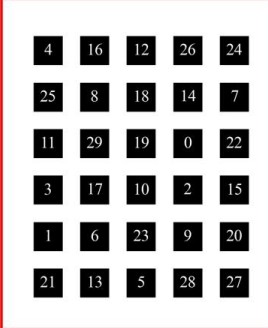

User: You are given a rectangular room with dimensions 227 cm by 278 cm.
The provided image represents the top-down view of the rectangular room with markers.
Each marker is associated with a coordinate (x, y) within the room. The markers are placed at a
↪ distance of 39 cm from each other.

Determine the mark that is exactly to the left of 29.
Ensure that the there exists indeed mark on. If no, please use [-1] to indicate it.

---

VLM Response : To determine the mark that is exactly to the left of mark 29, we need to understand
↪ the layout of the markers in the room.

Given:
- The room dimensions are 227 cm by 278 cm.
- Markers are placed 39 cm apart.
- The markers are arranged in a grid pattern.

First, let's calculate the number of markers along the width (227 cm) and the length (278 cm) of the
↪ room:

Number of markers along the width:

$$\text{Width markers} = \left\lfloor \frac{227}{39} \right\rfloor + 1 = \lfloor 5.82 \rfloor + 1 = 5 + 1 = 6$$

Number of markers along the length:

$$\text{Length markers} = \left\lfloor \frac{278}{39} \right\rfloor + 1 = \lfloor 7.13 \rfloor + 1 = 7 + 1 = 8$$

So, the grid is 6 markers wide and 8 markers long.

The markers are numbered sequentially from left to right, top to bottom. Therefore, the numbering
↪ would look like this:

```
1 2 3 4 5 6
7 8 9 10 11 12
13 14 15 16 17 18
19 20 21 22 23 24
25 26 27 28 29 30
31 32 33 34 35 36
37 38 39 40 41 42
43 44 45 46 47 48
```

To find the mark to the left of 29, we need to check if 29 is not in the first column. Since 29 is in
↪ the 5th row and 5th column, the mark to the left of 29 is 28.

Therefore, the mark that is exactly to the left of 29 is 28.

....

Answer: ["28"]

Figure 17: Illustration of a failure case for task S1 (Left) with the GPT-4o model using marks as the coordinate representation. We observed that the model's output was identical for both visual inputs (3D1) and sketches. Interestingly, in both scenarios, the model ignored the image and relied on the textual prior, incorrectly answering that the marker to the left of '29' is '28'.

```
User: You are given a rectangular room with dimensions 313 cm by 376 cm.
The provided image represents the top-down view of the rectangular room with markers.
Each marker is associated with a coordinate (x, y) within the room. The markers are placed at a
↪   distance of 53 cm from each other.

Determine the mark that is exactly to the right of 7.
Ensure that the there exists indeed mark on. If no, please use [-1] to indicate it.

---

VLM Response : To determine the mark that is exactly to the right of mark 7, we need to understand
↪   the layout of the markers in the room. Given that the markers are placed 53 cm apart, we can
↪   calculate the number of markers along the width (x-direction) and the length (y-direction) of the
↪   room.

1. **Calculate the number of markers along the width (x-direction):**

Number of markers along width = ⌊376/53⌋ + 1 = 7 + 1 = 8

2. **Calculate the number of markers along the length (y-direction):**

Number of markers along length = ⌊313/53⌋ + 1 = 5 + 1 = 6

So, there are 8 markers along the width and 6 markers along the length, forming a grid of 8 columns
↪   and 6 rows.

3. **Determine the position of mark 7:**
Assuming the markers are numbered row-wise starting from the top-left corner (0,0), the numbering
↪   would be as follows:
- Row 1: 1, 2, 3, 4, 5, 6, 7, 8
- Row 2: 9, 10, 11, 12, 13, 14, 15, 16
- and so on...

Mark 7 is in the first row, 7th column.

4. **Find the mark to the right of mark 7:**
The mark to the right of mark 7 would be in the same row, 8th column, which is mark 8.

Therefore, the mark that is exactly to the right of mark 7 is mark 8.

....

Answer: ["8"]
```

Figure 18: Illustration of a failure case for task S1 (Right) with the GPT-4o model using marks as the coordinate representation. The model ignored the image and relied on the textual prior, incorrectly answering that the marker to the right of '7' is '8'.