# OpenReview forum: "On Inherent 3D Reasoning of VLMs in Indoor Scene Layout Design"
_ICLR.cc/2025/Conference — Submitted to ICLR 2025_

### Official Review · Reviewer_H6Z8 · 2024-11-01

**Soundness:** 2
**Presentation:** 2
**Contribution:** 2
**Rating:** 3
**Confidence:** 5

**Summary:**

The paper systematically benchmarks current VLMs on various spatial reasoning tasks using layout, with an application in indoor scene synthesis. The results highlight several findings, including that these VLMs tend to prefer normalized coordinates and struggle to interpret complex visual representations. Additionally, the task heavily relies on language processing.

**Strengths:**

* Understanding and reasoning about indoor layouts is an important task with many valuable applications in various industries.
* The paper is well-written and easy to understand.

**Weaknesses:**

My main concern is about the motivation behind the tasks and the contribution of the key findings. The paper mentions that “Contrary to expectation, VLMs perform best with simplified sketch-based scene representations or, most strikingly, with no visual input at all, compared to detailed renderings.” However, this result isn’t entirely surprising. I think It raises questions about the choice of using VLMs for layout reasoning, as the tasks appear to be solvable using text alone, which aligns with the results showing better performance with text input/sketches. In that case, it seems more about testing the OCR capabilities of VLMs. Additionally, converting layouts to text appears to require less effort than rendering 3D. It would be helpful to clarify why VLMs are really necessary for these layout tasks.

My second concern is about the quantity and quality of the dataset. Having only 25 QAs per task seems insufficient. Additionally, it’s unclear whether steps were taken to ensure that the visual prompting is presented in a way that VLMs can recognize, for example, specific markers or axes designed for layouts. In some figures, the markers and numbers appear to be entirely occluded.

Finally, as already noted in the limitations, the tasks mainly focus on 2D reasoning rather than 3D, which makes the title somewhat misleading.

**Questions:**

* What is the motivation for using VLMs for layout reasoning?
* Would including axis and units as visual prompts be beneficial?
* It would be great to include human performance on each of the setups.

---

> ### Author Response · Authors · 2024-11-20
> **Response to Reviewer H6Z8**
>
> > **Q:** Motivation. It would be helpful to clarify why VLMs are really necessary for these layout tasks.
>
> **Response:** We believe VLMs are absolutely necessary for visual layout generation! We limit the scope of tasks here to simple rectangular bedrooms, where we are still able to demonstrate visual deficiencies in models. We include tasks where humans can trivially solve a task visually (S2 all-in) and find that VLMs still prefer symbolic computation over reasoning using the image. Our S3 task requires reasoning about affordances and functionality, which can change depending on the specific 3D assets under consideration. In the pursuit of creating a visual design assistant, many additional challenges will emerge that require vision, such as dealing with irregular room or object shapes – we currently only describe objects as 3D bounding boxes in the literature but we hope it’s easy to appreciate that they are much more than 3D boxes and offer complex affordances based on their shape. The shape of the arm of a sofa decides whether a human could sit on it. There exist objects that do not have a specific orientation defined by functionality such as plants, carpets/rugs and certain kinds of lights being a few examples. Of course, one could argue that detailed descriptions of these could suffice, which would be equivalent to arguing that a blind human could easily approach design with detailed descriptions.
> We attribute the higher performance of VLMs with text and simplified sketches to the following, which we believe uncovers deficiencies in the data and methods for training VLMs:
> - Bias towards symbolic computation when clear visual evidence is available
> - Reduction in symbolic reasoning performance in the presence of image input
>
> **Q:*** Quantity and quality of the dataset.
>
> While we use 25 QA pairs per sub-task (per question and rendering type), they overall constitute 3400 QA pairs over all our tasks which is comparable or larger than datasets used for evaluating VLMs [1,2,3]. The cost of evaluating VLMs is a concern towards increasing the size of evaluation datasets and the community has evolved to prefer many small eval datasets for holistic evaluation of diverse skills in VLMs. Moreover, we ensure that for every question, information relevant to the question is always available either in text or visible on the image. For example, in S3, where most occlusion is possible, the target marker when a new object is to be placed is always visible from the rendered viewpoint.
>
> [1] Eyes wide shut? exploring the visual shortcomings of multimodal llms. Tong et al.
> [2] Cambrian-1: A Fully Open, Vision-Centric Exploration of Multimodal LLMs. Tong et al.
> [3] WildVision: Evaluating Vision-Language Models in the Wild with Human Preferences, Lu et al.
>
> **Q:** Finally, as already noted in the limitations, the tasks mainly focus on 2D reasoning rather than 3D, which makes the title somewhat misleading.
>
> **Response:** We inherit this 2D reasoning for the 3D indoor scene synthesis task from the indoor scene synthesis literature which generally follows this abstraction [1,2,3,4]. Since all reviewers mention this confusion, we will remove 3D from our title to prevent further misunderstanding. We believe our findings are directly applicable to current indoor scene synthesis methods and hope that this satisfies the reviewer’s concern.
>
> [1] Holodeck: Language Guided Generation of 3D Embodied AI Environments, Yang et al. (CVPR 2024)
> [2] DiffuScene: Denoising Diffusion Models for Generative Indoor Scene Synthesis, Tang et al. (CVPR 2024)
> [3] ATISS: Autoregressive Transformers for Indoor Scene Synthesis, Paschalidou et al. (NeurIPS 2021)
> [4] Fast and Flexible Indoor Scene Synthesis via Deep Convolutional Generative Models, Ritchie et al. (CVPR 2019)
>
> > **Q:** Would including axis and units as visual prompts be beneficial?
>
> **Response:** Indeed, including an axis and unit as visual prompt could be beneficial. There is a submission to ICLR which explores this exact question (https://openreview.net/forum?id=IXOoltTofP)
> It would be great to include human performance on each of the setups.
> We agree with the reviewer on this! We note that our S1 and S2 task are trivially solvable by humans. Our S3 task is more difficult and we will study human performance on this task. Note that the dataset is created from human generated designs from the 3D-Front dataset and human created negatives, and all VLMs show chance performance on S3.
>
> > **Q:** It would be great to include human performance on each of the setups.
>
> **Response** We agree with the reviewer on this! We note that our S1 and S2 task are trivially solvable by humans. Our S3 task is more difficult and we will study human performance on this task. Note that the dataset is created from human generated designs from the 3D-Front dataset and human created negatives, and all VLMs show chance performance on S3.
>
> We thank the reviewer for their time and look forward to continued discussion!

---

> > ### Comment · Reviewer_H6Z8 · 2024-11-25
> >
> > Thank you for your response. However, I have several concerns that still seem unresolved:
> >
> > **Necessity of VLMs for layout tasks.** While I agree with the point that *“The shape of the arm of a sofa decides whether a human could sit on it…”*, I didn’t see scenarios like this reflected in the current paper (please correct me if I’m mistaken). This makes it a bit unclear why 2D VLMs are particularly necessary compared to alternatives like text-based LLMs or 3D LLMs.
> >
> > **Quantity and quality of the dataset.** The clarification on dataset quantity was helpful, but I still have some concerns about quality. For example, in Figure 4, some tags appear to be occluded.
> >
> > **Human performance.** I agree that S1 and S2 tasks are trivially solvable by humans. However, while the authors mentioned plans to study human performance on S3, I couldn’t find this addressed in the response or the revised paper.

---

### Official Review · Reviewer_3xqr · 2024-11-02

**Soundness:** 2
**Presentation:** 1
**Contribution:** 1
**Rating:** 3
**Confidence:** 3

**Summary:**

This paper systematically studies the ability of off-the-shelf VLM to reason about spatial relations. The authors design problems from 3 aspects, namely coordinate communication, free-space reasoning and complex joint reasoning to evaluate the performance of the VLM. Experiments show that VLM struggles at specific cases like absolute coordinates or markers as well as inherent error for reasoning as in the free-space understanding case.

**Strengths:**

1. The paper attempts to systematically evaluate VLM's ability to understand spatial relations rather than designing a pipeline for a specific case.
2. Extensive experiments are conducted to derive systematic conclusions.

**Weaknesses:**

1. The major issue with this work is the overclaiming of evaluating VLM for 3D reasoning while the efforts are all made on 2D.
2. Since the efforts are all made on 2D, it is worth noticing that the findings in the paper are mostly well-studied in the previous literatures[1][2][3].
3. The paper is poorly written, with bunch of typos in the main text and tables, making it hard to follow.


[1] Favero, Alessandro, et al. "Multi-modal hallucination control by visual information grounding." Proceedings of the IEEE/CVF Conference on Computer Vision and Pattern Recognition. 2024.

[2] Rahmanzadehgervi, Pooyan, et al. "Vision language models are blind." arXiv preprint arXiv:2407.06581 (2024).

[3]  Huang, Qidong, et al. "Opera: Alleviating hallucination in multi-modal large language models via over-trust penalty and retrospection-allocation." Proceedings of the IEEE/CVF Conference on Computer Vision and Pattern Recognition. 2024.

**Questions:**

1. Since the paper claims to study the problem in the 3D domain, it is suggested that the evaluation of the model should focus more on how to provide the right 3D representations or how to formulate the questions. For example, different angles of the rendered scene with no pre-defined coordinate systems to ask questions about spatial relations like left and right.
2. It seems like the paper is solely studying VLM's performance in a limited 2D domain, it would be beneficial if the authors could provide a clarification on how the task is different from or harder than assessing VLM on unlimited domains of images.

Other issues:
1. Tab. 2 missing the final Avg. score. The caption is not sufficient for readers to understand the score marked as red and bolded.
2. "an With, ..." in Ln. 297, I assume "an " should not be there.
3. Inconsistent formatting of Table. Tab. 5 does not have bolded text and red text, and Tab. 3 does not have red text.


I am open to discussion if I misunderstood the intention or content of the paper.

---

> ### Author Response · Authors · 2024-11-20
> **Response to Reviewer 3xqr**
>
> > **Q:** The major issue with this work is the overclaiming of evaluating VLM for 3D reasoning while the efforts are all made on 2D.
>
> **Response:** We agree with the reviewer and also discuss in our limitations section that our tasks could be considered 2D reasoning. This generally reflects the state of the 3D indoor scene synthesis field [1,2,3,4], which most commonly abstracts the problem into the placement of furniture on a floorplan. We did intend to extend this to layered furniture, but found that the floorplan placement task is already challenging for VLMs and hence limited our scope. We believe our findings apply directly to the 3D indoor scene synthesis literature. To avoid any further confusion, we will remove 3D from our title.
>
> [1] Holodeck: Language Guided Generation of 3D Embodied AI Environments, Yang et al. (CVPR 2024)
>
> [2] DiffuScene: Denoising Diffusion Models for Generative Indoor Scene Synthesis, Tang et al. (CVPR 2024)
>
> [3] ATISS: Autoregressive Transformers for Indoor Scene Synthesis, Paschalidou et al. (NeurIPS 2021)
>
> [4] Fast and Flexible Indoor Scene Synthesis via Deep Convolutional Generative Models, Ritchie et al. (CVPR 2019)
>
> > **Q:** Since the efforts are all made on 2D, it is worth noticing that the findings in the paper are mostly well-studied in the previous literatures
>
> **Response:** Thank you for pointing out papers on visual information grounding. These are indeed relevant and valuable techniques to improve decoding with better visual grounding. We will include these in our discussion as potential methods to improve performance on our tasks. We believe our findings supplement the findings of concurrent work of Pooyan et al. (Vision language models are blind) since we test models with text-only, simplified 2D sketches as well as 3D rendering of indoor rooms, from multiple view points in a unified way. Moreover, we test methods to communicate coordinates with VLMs whereas prior work has used different methods without standardization. Furthermore, our tasks are clear decompositions of primitive skills required for indoor scene synthesis, with our last experiment also including understanding affordances and object functionality. Since VLMs already struggle at these tasks, we limited our scope and did not go beyond to include layered furniture (such as stuff on a table, or a chair pushed under a table which is not discernable from 2D alone) which would require additional 3D understanding skills.
>
> > **Q:** Since the paper claims to study the problem in the 3D domain, it is suggested that the evaluation of the model should focus more on how to provide the right 3D representations or how to formulate the questions. For example, different angles of the rendered scene with no pre-defined coordinate systems to ask questions about spatial relations like left and right.
>
> **Response:** We note that while qualitative relationships such as left or right are studied in prior work such as Visual Spatial Reasoning, Liu et al. ACL 2023. Since the task of 3D scene synthesis also requires precise placement and assessment of furniture, we focus on dealing with coordinates over qualitative relationships. We would also like to restate that the field of 3D indoor scene synthesis currently focuses on layout synthesis on a floor-plan. As discussed in the paper, prior state of the art work on LLM-based indoor scene synthesis such as Holodeck (Yang et al.) has focused on generating “object relationship graphs” in a scene which are converted to precise placements using a heuristic algorithm. We ask whether a VLM could be used to replace the heuristic algorithm to make an end-to-end indoor scene synthesis method with LLMs and VLMs. We will be removing 3D from the title of our paper to avoid further confusion. Our tasks are all solvable from a single image alone by humans and hence we do not test 3D representations as input to models specifically trained to work with 3D inputs such as camera information with multi-view images or point clouds. However, we agree that these would be interesting future work and since our data is synthetically generated, future users could modify our codebase to perform such research.

---

> ### Author Response · Authors · 2024-11-20
> **Continued response to Reviewer 3xqr**
>
> >**Q:** It seems like the paper is solely studying VLM's performance in a limited 2D domain, it would be beneficial if the authors could provide a clarification on how the task is different from or harder than assessing VLM on unlimited domains of images.
>
> **Response:** While the indoor scene synthesis task is indeed abstracted into 2D floorplan placement in current literature, real world applications would necessitate complete 3D understanding from renderings. Our finding that simplified 2D sketches are preferred to 3D rendering in our tasks shows that current gen VLMs are not ready to be used as general purpose visual design assistants due to their failure on simplistic tasks. Since our evaluation is on controlled rendered data, we can test models with the exact same data in different configurations, such as our variation of rendering viewpoints, or whether different objects in question are rendered or not. For example, for a VLM to understand whether a target object can be placed at a specified location, if that target object needs to be rendered (S2 all-in), this implies an O(N) rendering steps which is expensive, vs. if the VLM can make the judgement while only rendering existing objects or none at all (S2 existing-in, S2 Empty) which is O(1) in rendering steps. Understanding these are important to building a visual design assistant and we believe our evaluation has simplicity while being difficult for current-gen VLMs and being applicable to the design of a LLM and VLM based visual design assistant. Generating such data and insights from natural images would be difficult, but we do agree that if possible, that would enable evaluation at a much large scale and diversity. We will modify the paper to highlight this better and we thank the reviewer for the suggestion.
>
> > **Q:** Tab. 2 missing the final Avg. score. The caption is not sufficient for readers to understand the score marked as red and bolded.
>
> **Response:** We apologize for this oversight. We will remove the red text as it was an error on our part. We use bold text to show the best performance, we will improve the caption to reflect this. We omit the final Avg. score since that would average over all models and all question types, which we do not believe is valuable information. The averages per model reflect the strength of each model, whereas the averages over All-in, Empty and Existing-in show the strength of question types, which shows us that the “Empty” question type shows superior performance across models and corroborates our finding that VLMs do better with lesser (best with none) image information in these tasks, which we believe is due to:
> - Over-reliance on symbolic processing in the presence of visual information, and
> - Degradation in symbolic reasoning abilities in the presence of visual information
>
> > **Q:** "an With, ..." in Ln. 297, I assume "an " should not be there.
>
> **Response** Thank you for noticing this. We will fix this.
>
> > **Q:** Inconsistent formatting of Table. Tab. 5 does not have bolded text and red text, and Tab. 3 does not have red text.
>
> **Response** Indeed, including red text in previous tables was an oversight and we will correct them. In Table 5, we do not show any bold text since all models have close to chance performance and are hence not worth highlighting. Once again, we will improve the text to reflect this reasoning. Please let us know if the reviewer agrees with these proposed changes.
>
> We thank the reviewer for their time and valuable comments and look forward to continued discussion!

---

> > ### Comment · Reviewer_3xqr · 2024-11-27
> >
> > Thank you for your detailed responses and I appreciate the point-by-point clarification the authors made. However, my concern on 3D and 2D domain still remains. Specifically, the effort in the paper is mostly made on 2D images in limited domain, constraining the evaluation of MLLM to a restricted field. Therefore, the discoveries/conclusions in the paper are well-covered in relevant literature[1, 2, 3, 4, 5]. I do find the idea of testing MLLM's ability on 3D reasoning and on 3D indoor scene design interesting. I suggest the author extend the models studied from only taking 2D input to video reasoning systems / 3D point cloud reasoning systems, which could make the study more insightful and convincable.
> >
> >
> > [1] Chen, Boyuan, et al. "Spatialvlm: Endowing vision-language models with spatial reasoning capabilities." Proceedings of the IEEE/CVF Conference on Computer Vision and Pattern Recognition. 2024.
> >
> > [2] Cheng, An-Chieh, et al. "SpatialRGPT: Grounded Spatial Reasoning in Vision Language Model." arXiv preprint arXiv:2406.01584 (2024).
> >
> > [3] Favero, Alessandro, et al. "Multi-modal hallucination control by visual information grounding." Proceedings of the IEEE/CVF Conference on Computer Vision and Pattern Recognition. 2024.
> >
> > [4] Rahmanzadehgervi, Pooyan, et al. "Vision language models are blind." arXiv preprint arXiv:2407.06581 (2024).
> >
> > [5] Huang, Qidong, et al. "Opera: Alleviating hallucination in multi-modal large language models via over-trust penalty and retrospection-allocation." Proceedings of the IEEE/CVF Conference on Computer Vision and Pattern Recognition. 2024.

---

### Official Review · Reviewer_tM3P · 2024-11-03

**Soundness:** 2
**Presentation:** 2
**Contribution:** 2
**Rating:** 5
**Confidence:** 4

**Summary:**

This work designs three categories of 3D reasoning question-answering tasks situated in simulated bedrooms to evaluate Vision Language Models (VLMs) on their capabilities as potential assistants for indoor scene layout design. The authors designed questions in different types and different input modalities. Their experiments benchmarked a variety of commonly used proprietary and open-source VLMs. Their findings indicate that VLMs generally struggle to answer spatial-reasoning questions (type S2 and S3 in their paper) based on visual inputs and tend to rely only on text input. They also find that the VLMs can reason normalized coordinates much better than absolute coordinates and simple grid-like coordinate marks. The authors hope this benchmark and their discoveries can benefit the development of next-generation VLMs.

**Strengths:**

1) Accessing the spatial reasoning capabilities of VLMs has recently become an important direction, and this paper approaches it from the perspective of indoor scene layout design, which is a kind of novelty given that most of the literature in this area is very recent.

2) The experiments are extensive. The authors incorporated 3400 questions across various types and tested five modalities for five state-of-the-art VLMs ranging from proprietary to public models.

**Weaknesses:**

1. Lack of in-depth analysis. While the authors' efforts in exposing the limitations of the current VLMs are appreciated, it would grant the paper more impact on the community if they could further analyze possible reasons for these models' failures in 3D reasoning and even design new methods to try to mitigate these failures. [1], which is also referred to in the paper, is a great example. This will help the paper better meet the expectations of a top-tier conference like ICLR. Currently, the paper only indicates "underscore the need for improved training methodologies and data curation" in the conclusion without elaboration.

2. Inclusion of 3D LLMs. While the paper acknowledges this as a limitation, given the context of the tasks (3D reasoning / spatial reasoning), the reviewer believes it is important to include 3D LLMs in the experiments.

3. Diversity in room appearances. While details about the prompts and multiple images are provided in the paper, all the rooms seem to have the same appearance (square shape, wooden floor, and light brown walls). Including more shapes, colors, and textures in the dataset could improve the quality, realism, and generalizability of the dataset.

4. Embodied view is skipped for task type 3, limiting the experiment’s design and findings to scenarios where top-down or perspective views are available, such as indoor scene layout design. This reduces applicability to broader spatial reasoning tasks in embodied AI, where these views may not be accessible.

---

[1] Tong, Shengbang, et al. "Eyes wide shut? exploring the visual shortcomings of multimodal llms." Proceedings of the IEEE/CVF Conference on Computer Vision and Pattern Recognition. 2024.

**Questions:**

1. Different prompting techniques can impact VLM performance. From the paper, it appears that the wording for specific question types remains the same. Have you tried using chain-of-thought prompting or applying visual prompts to the input images? Are there alternative phrasings for the questions in the dataset?

2. How should people leverage these findings from this paper in training a VLM with better spatial reasoning ability?

3. other minor issues:

> line 151 combaining -> combining
>
> line 297, please revise. The sentence contains obvious grammar mistakes.

---

> ### Author Response · Authors · 2024-11-20
> **Response to Reviewer tM3P**
>
> > **Q:** Lack of in-depth analysis.
>
> **Response:** We agree with the reviewer that understanding possible reasons for sub-par performance on our evaluation is necessary to improve the next generation of VLMs. Simultaneously, to measure progress on a skill in VLMs, we require evaluation data. While LLM-based indoor scene synthesis methods seem to produce excellent results (such as Holodeck, Yang et al. CVPR 2024), they might also signal a false sense of spatial understanding in LLMs. In their specific case, object placement is done externally from the LLM, through an iterative backtracking solver that exhaustively searches for potential solutions to a generated constraint graph. Our evaluation is intentionally simple and breaks down the indoor scene synthesis task into easily evaluated core components and shows limitations in problems that could be trivially solved by humans (S1 and S2). We believe this raises valuable awareness of the true spatial capabilities of current VLMs for the indoor scene synthesis task (and LLMs since we evaluate text-only performance as well). Moreover, we believe three findings merit targeted exploration beyond the indoor scene synthesis task:
> - Bias towards symbolic computation when clear visual evidence is available
> - Reduction in symbolic reasoning performance in the presence of image input
> - The availability of functions to compute free-space over perceiving it can readily improve performance on S2, which can be used for new methods using current-gen LLMs and VLMs for spatial tasks
>
> > **Q:** Inclusion of 3D LLMs.
>
> **Response:** While 3D LLMs would be a good candidate for such tasks, our tasks are completely solvable by a human from a single image. 3D LLMs that we are aware of (and are available to test) require additional inputs such as camera info, depth, multi-view images, point clouds. We could produce this data and agree with the reviewer that a future version of the benchmark could include such information to allow testing 3D LLMs as well. Our evaluation is intentionally simple and can be solved from a single image by humans, sometimes trivially, such as our S2 all-in task. We believe this builds valuable awareness of lack of spatial reasoning in current image-only VLMs. We believe it will be valuable to add this discussion to our manuscript and thank the reviewer for the suggestion. Moreover, if the reviewer has a particular 3D LLM in mind that is compatible with our scope i.e. single images and is available for testing, we would appreciate a concrete reference and we will be happy to experiment and add results.
>
> > **Q:** Diversity in room appearances.
>
> **Response:** Our findings in-fact show that simple visual rendering (sketches) induce better reasoning performance in VLMs over 3D rendering. Hence we believe that improved and diverse rendering in the dataset would add complexity to the task that is currently unnecessary. As models evolve and saturate this benchmark, such diversity as proposed by the reviewer could become useful. Additionally, we intentionally re-render the data ourselves instead of using prior 3D-front rendering methods (such as that from the original dataset or from ATISS, Paschalidou et al. which is commonly used) to ensure the data is not seen in training. We will discuss this in our updated manuscript.
>
> > **Q:** Embodied view is skipped for task type 3
>
> **Response:** This is intentionally done since a single embodied view of the room might not see the whole room context. To simplify the tasks to be a single image task (not all VLMs support multiple images), we skip the embodied view. We agree that this does limit the applicability of S3 to broader spatial reasoning tasks in embodied AI. We will release our rendering code, which future users could use to render multiple views or other 3D information to test on the same task.

---

> ### Author Response · Authors · 2024-11-20
> **Continued response to Reviewer tM3P**
>
> >**Q:** Different prompting techniques can impact VLM performance. Have you tried using chain-of-thought prompting or applying visual prompts to the input images?
>
> **Response:** Indeed. We tried multiple phrasings and started from prompts used by Layout-GPT (Feng et al.) and Holodeck (Yang et al.). We also tried CoT prompting and do test with Set-of-marks visual prompting in the paper. Using CoT resulted in no performance change, and we report a subset of the performance here on our S2 task with gpt-4o-2024-05-13 (used in the paper) and the newer gpt-4o-2024-08-06. We report the results below, and we will add a discussion regarding this to our supplementary material and refer to it in the main paper.
>
> | Model                        | all_in | empty  | existing_in | Grand Total |
> |------------------------------|--------|--------|-------------|-------------|
> | gpt-4o-2024-05-13      | 0.738  | 0.766  | 0.718       | 0.742       |
> | gpt-4o-2024-05-13-CoT  | 0.688  | 0.724  | 0.713       | 0.709       |
> | gpt-4o-2024-08-06      | 0.778  | 0.817  | 0.795       | 0.798       |
> | gpt-4o-2024-08-06-CoT  | 0.753  | 0.785  | 0.763       | 0.768       |
>
> >**Q:** How should people leverage these findings from this paper in training a VLM with better spatial reasoning ability?
>
> **Response:** Along with providing a benchmark to measure spatial performance in our task, we believe there are three actionable findings that can be studied independently. While these are not techniques to readily improve training, we believe they can be used to inform the kind of data utilized in pre-training or post-training / visual instruction tuning.
> - Bias towards symbolic computation when clear visual evidence is available
> - Reduction in symbolic reasoning performance in the presence of image input
> - The availability of functions to compute free-space over perceiving it can readily improve performance on S2, which can be used for new methods using current-gen LLMs and VLMs for spatial tasks
>
> >**Q:** other minor issues:
>
> **Response:** Thank you for bringing these to our attention. We will fix these in our revision.
>
> We thank the reviewer for their time and valuable comments and look forward to continued discussion!

---

> > ### Comment · Reviewer_tM3P · 2024-11-25
> >
> > Dear authors,
> >
> > Thank you for your response and I appreciate the added CoT experiments.
> >
> > However,  I feel several of my concerns still remain valid:
> >
> > **For the inclusion of 3D LLMs.** I agree with the authors' explanation that the proposed tasks are solvable by humans from single image observation. However, as mentioned by other reviewers also, this paper is situated as a paper for **3D** spatial understanding, so I still feel it would make the paper less aligned with the claim if the benchmark is involved with just 2D. After all, the scene is simulated and multiple views are available, so essentially you will have access to all kinds of 3D metric information such as camera poses, depths, 3D object bounding boxes, and etc. I am not an expert in house design and construction but I feel these pieces of information are useful and obtainable in reality as well. Therefore, if the paper is targeted at real applications, I still believe it is necessary to include 3D LLMs. On the other hand, if the paper is instead targeted at analyzing and understanding MLLMs, then I agree with the other reviewers that the scope and contribution are less significant compared to the relevant literature.
> >
> > **For the data diversity.** If simple sketches are the best way of input, then why not try to use real floor plans and build your benchmark on that? Floor plans are clearly available online in abundant amounts, and in various layouts and sizes. The reviewer believes they will greatly increase your benchmark's diversity (better than just a square-shaped room) and make the task more convincing by being better situated in real-life applications.  Do the authors think this is a good suggestion?
> >
> > In summary, I would like to keep my original rating. I will also keep track of the other reviewers' feedback and stay engaged in the discussion.

---

### Official Review · Reviewer_dLVZ · 2024-11-03

**Soundness:** 2
**Presentation:** 2
**Contribution:** 2
**Rating:** 5
**Confidence:** 4

**Summary:**

This paper evaluates large vision-language models (VLMs) for their inherent 3D reasoning abilities in the context of indoor scene layout tasks, focusing on key spatial tasks like locating objects, reasoning about free space, and managing object alignment. Findings reveal that VLMs are often better at symbolic reasoning than at directly utilizing visual inputs, and they struggle with complex spatial tasks, such as preventing collisions and ensuring functional alignment in room layouts.

**Strengths:**

1. Task Decomposition. The study clearly breaks down scene reasoning into essential sub-tasks, making the evaluation framework easy to follow and understand.

2. Insight into VLM Shortcomings. The paper provides useful insights into the current limitations of VLMs, especially their over-reliance on language-based rather than visual-based reasoning, which highlights areas for further model improvement.

**Weaknesses:**

1. Unclear Positioning. The paper’s approach to 3D generation is somewhat ambiguous, as it abstracts the problem into a simplified 2D scene layout task. This positioning may limit the relevance of the findings to true 3D reasoning, as spatial complexity is reduced to single-plane reasoning rather than robust 3D spatial understanding.

2. Limited Dataset and Task Scope. The study focuses on a single-room setup using standardized indoor layouts and simplified tasks based on the 3D-FRONT dataset. This narrow scope fails to represent real-world 3D complexities, such as handling overlapping or interleaved objects, which are common in actual environments. For example, scenarios involving stacked furniture or layered decor, where objects partially obscure each other, are not addressed, limiting the study’s applicability to more realistic 3D spatial tasks.

3. Over-Reliance on Symbolic Reasoning. VLMs in this study default to symbolic rather than visual reasoning. This reliance raises concerns about their effectiveness in spatially complex tasks that require direct interpretation of visual inputs, thus questioning the applicability of the findings to tasks that need strong visual comprehension.

4. Single Data Source. The evaluation relies heavily on the 3D-FRONT dataset and does not incorporate diverse datasets that could better assess generalization. A wider range of data sources covering various spatial configurations and room types would provide a more comprehensive measure of VLMs’ 3D reasoning capabilities.

5. Incomplete Model Evaluation. The paper’s contribution focuses on defining new tasks and evaluating existing models, yet it lacks a broader comparison with other mainstream VLMs. A more thorough evaluation across a wider variety of models, potentially with an online leaderboard, would enhance the paper’s contributions by establishing a clearer benchmark for future research.

**Questions:**

1. Free-Space Reasoning with Minimal Visual Inputs. The study notes that VLMs often perform better with minimal or no visual inputs. Can the authors elaborate on why simplified visual representations or text-only prompts might lead to improved accuracy, and how this insight might influence future VLM design?

2. Evaluation Strategy for Task Complexity. For tasks where symbolic reasoning seems to dominate, have the authors considered introducing hybrid evaluation strategies that encourage models to rely more on visual inputs? Could certain prompt engineering or visual cues shift the balance towards true visual reasoning?

---

> ### Author Response · Authors · 2024-11-20
> **Response to Reviewer dLVZ**
>
> > **Q**: Unclear Positioning.
>
> **Response**: The reviewer is right and we use this abstraction following representative indoor scene layout synthesis methods in the literature [1, 2, 3, 4]. We agree that this does not reflect in-the-wild 3D reasoning which will require further work in both 3D layout synthesis methods and in evaluation, where the latter is our focus. We will update our manuscript to clarify this distinction. Moreover, to avoid confusion, we will remove 3D from the paper title.
>
> [1] Holodeck: Language Guided Generation of 3D Embodied AI Environments, Yang et al. (CVPR 2024)
>
> [2] DiffuScene: Denoising Diffusion Models for Generative Indoor Scene Synthesis, Tang et al. (CVPR 2024)
>
> [3] ATISS: Autoregressive Transformers for Indoor Scene Synthesis, Paschalidou et al. (NeurIPS 2021)
>
> [4] Fast and Flexible Indoor Scene Synthesis via Deep Convolutional Generative Models, Ritchie et al. (CVPR 2019)
>
>
> > **Q**: Limited Dataset and Task Scope.
>
> **Response**: This is a valid criticism, and again reflects indoor scene synthesis literature as a whole, where most methods work on similar abstractions. Our original intention was to extend to stacked furniture (stuff on a table, for example.). However, we find that VLMs already struggle at understanding furniture placement on a 2D floorplan, which is why we limited our scope of evaluation to make it concise. Moreover, large public datasets with stacked furniture / layered decor are yet to be made available to enable such research.
>
> > **Q**: Over-Reliance on Symbolic Reasoning.
>
> **Response**:
> - We indeed observe this. Our final evaluation experiment (S3) specifically addresses this question. In S3, we do not provide text inputs that could enable symbolic reasoning, thus forcing the model to visual reasoning alone. We find that all models we test have close to chance performance. In our S2 all-in task, where both the target and existing furniture are rendered, the task is trivially solvable visually. Yet, the VLMs we evaluate chose to reason symbolically. This is indeed a clear limitation of VLMs on a simple visual task, and our attempts to alleviate this with prompting to focus on the image did not help. Our intention is thus to leave this as a problem for the community to solve through better prompting, or through better training for visual reasoning, where symbolic reasoning is not the preferred mode.
> - The other two evaluations, S1 and S2, provide both symbolic and visual input, where the VLM is free to choose any or both. This enables us to fairly evaluate models in text-only mode i.e. without any visual input. This enabled us to find that the addition of visual inputs decreases symbolic reasoning performance in current VLMs.
>
>
> > **Q**: Single Data Source.
>
> **Response**: The 3D-FRONT dataset does contain additional room types and furniture that we could add with our data generation script, which we intend to release. We focus on bedrooms with rectangular layouts following layout-gpt (Feng et al.), since it allows us to simply define room boundaries in text to evaluate in text-only mode. We intentionally re-render 3D-Front in a novel style with Blender to ensure the data is not seen during training and hence evaluates generalization to some degree. We agree that a future version of such an evaluation dataset (once this eval is saturated) should include diverse room types, layered furniture and diverse styles to capture more of the real world. For now, we intentionally scope our data to be as simple as possible while still showing stark limitations in current models.

---

> > ### Author Response · Authors · 2024-11-20
> > **Continued response to Reviewer dLVZ**
> >
> > > **Q**: Incomplete Model Evaluation.
> >
> > **Response**: We focused our evaluation of GPT-4o and GPT-4o-mini as frontier API-access models and evaluated LLAMA-3.2-90B, LLaVA-Next-110B and Qwen-VL-72B as frontier open access models. We agree that we could evaluate Claude and Gemini additionally for frontier API-access models and Molmo, as a frontier open access mode. We did not evaluate VLMs fine-tuned for spatial tasks such as grounding or pointing since we intend to specifically evaluate general purpose VLMs that could function as a part of a visual design assistant performing an array of tasks of which our current evaluation is only a small part. We will add further discussion regarding this in the paper. For the rebuttal, we additionally ran evaluation for Molmo-72B-0924, Claude-3.5-Sonnet-20241022 and gpt-4o-2024-08-06 (newer version of 4o, we used gpt-4o-2024-05-13 in the paper) and will include them in the paper.
> >
> > Results for Claude and Molmo, and gpt-4o-2024-08-06 for comparison on our S2 task (S3 has chance performance as other models) are below. We notice that the new gpt-4o has improved on our S2 task by ~5% on average while claude-3.5-sonnet is close behind. Yet, both models still do best on this task with least visual information and worse with more visual information being made available to them. Moreover, both models still perform best without any image input (i.e. text only) with gpt-4o-2024-08-06 reaching 0.92 accuracy and claude-sonnet-3.5 reaching 0.84 accuracy compared to gpt-4o-2024-05-13's 0.85 reported in the paper.
> >
> > | Model                         | all_in | empty  | existing_in | Grand Total |
> > |-------------------------------|--------|--------|-------------|-------------|
> > | allenai_Molmo-72B-0924  | 0.471  | 0.576  | 0.493       | 0.517       |
> > | claude-3.5-sonnet-20241022 | 0.745  | 0.811  | 0.783       | 0.781       |
> > | gpt-4o-2024-08-06       | 0.778  | 0.817  | 0.795       | 0.798       |
> >
> > > **Q:** Free-Space Reasoning with Minimal Visual Inputs.
> >
> > **Response**: Since some of the data used to train models and their architectures are not always available publicly, we can only speculate. We speculate that models are trained for reasoning on datasets with diagrams (such as mathematical reasoning) which might explain better performance on diagram-like sketch inputs over 3D inputs. In Fig. 10 and Fig. 11 in our supplementary, we show an example where GPT-4o follows a very similar reasoning path with and without image inputs. We thoroughly verify that this is not a bug and that the model is able to see the image through additional prompting to describe the image. This indicates that when performing symbolic reasoning, the model might ignore the input image completely.
> > We could speculate a few things about VLMs here. However, since these are only speculations without experimental validation, we refrained from such discussion in the paper.
> >
> > - Lack of abundant and diverse visual reasoning training data [1]. A big part of the visual datasets used for pre/post training of open-source VLMs aim to improve image captioning abilities, over more complex visual reasoning. Raising awareness of VLMs lacunae in such tasks could lead to better data being used in future work. For example, we observed in the literature that GPT-4V performed poorly on tasks in SpatialVLM [2]. In a later paper [3] it was found that GPT-4o’s performance on similar tasks was dramatically improved, which we could speculate was due to the awareness raised by [1] to improve training data.
> > - Visual reasoning on mathematical tasks might only need limited visual reasoning before defaulting to symbolic manipulation, which is what we observed. We speculate a big component of the reasoning capabilities on VLMs might come from the capability of the underlying LLM to solve problems symbolically. Again, we believe this could be alleviated with more diverse visual reasoning data, perhaps including data similar to our evaluation data.
> >
> > On our tasks where we can evaluate fairly without visual inputs (S1 and S2), we observe that symbolic computation abilities of VLMs degrade on average when given visual inputs, as shown in our quantitative results. We believe this is a clear finding with experimental validation and should be actionable towards building the next generation of VLMs.
> >
> > [1] Cambrian-1: A Fully Open, Vision-Centric Exploration of Multimodal LLMs, Tong et al.
> >
> > [2] Spatial VLM: Endowing Vision-Language Models with Spatial Reasoning Capabilities, Chen et al.
> >
> > [3] Reasoning Paths with Reference Objects Elicit Quantitative Spatial Reasoning in Large Vision-Language Models, Liao et al.

---

> ### Author Response · Authors · 2024-11-20
> **Continued response to Reviewer dLVZ**
>
> > **Q:** Evaluation Strategy for Task Complexity.
>
> **Response:** This is a great suggestion and we could extend evaluating S1 and S2 without providing the necessary text input to be able to perform symbolic reasoning. Our current evaluation helps us evaluate fairly across visual modalities and without any vision input by providing the same prompt to the model. When an image is provided, we only add the following prompt – “Please use the attached image for reference. Both existing and target objects are shown in the image”. Our experiments with additional prompting, including prompting to first describe the image and then answer the question failed. We did not include them in the paper to keep the scope and evaluation contained, but this could indeed be a future strategy to improve performance on such data.
>
> We thank the reviewer for their time and valuable comments and look forward to continued discussion!

---

### Meta-Review · Area_Chair_fNCq · 2024-12-08

**Metareview:**

**Summary**

This paper investigates the ability of VLMs to design 3D indoor scenes.  Specifically the ability of VLMs to communicate about spatial locations (S1), reason about free space and object collisions (S2), and reason about object alignment, orientation, and functionality when placing a object in a room (S3).  Several different VLMs are compared. Findings indicate that VLMs are still poor at spatial reasoning, and relying purely on VLMs does not create reasonable scene layouts.

**Strengths**
Reviewers appreciated the following aspects of the work:
1. Evaluating spatial reasoning capabilities of VLMs is important [tM3P,H6Z8]
2. Clear task decomposition and evaluation [dLVZ]
3. Extensive experiments [tM3P,3xqr]
4. Some insights into shortcomings of VLMs [dLVZ]
5. Some reviewers find the paper to be well-written and easy to understand [H6Z8]

**Weaknesses**

The main concern expressed by all reviewers is whether this work is actually tackling 3D spatial understanding or just 2D.  Reviewers also felt that despite the experiments conducted, the overall evaluation is limited and not provide deep insights into why the VLMs performs poorly.

Concerns expressed by reviewers include:
1. Problem is simplified to be 2D scene layout [dLVZ,3xqr,H6Z8] which is misleading.  It is not clear how much 3D reasoning the VLM actually need to do.
2. Limited dataset and scope [dLVZ,tM3P,H6Z8]
3. Limited evaluation and lack of in-depth analysis.  Some directions identified by reviewers are:
   - No study of how to provide the right 3D representation or how to formulate the question [3xqr]
   - Impact of different prompting strategies [tM3P]
   - Only limited set of VLMs are evaluated [dLVZ]
   - Recent 3D-LLMs not included [tM3P]
   - Lack of human performance for comparison [H6Z8]
4. Over-reliance on symbolic reasoning, vs checking visual reasoning [dLVZ]
5. Some reviewers also pointed out issues with some of the tables and various typos [tM3P, 3xqr]

**Recommendation**

Based on reviewer scores and comments, the AC feels that the work is not appropriate in its current form for presentation at ICLR.  It is possible that the work can be of interest to a more targeted workshop or reworked to be clearer in its positioning.

**Additional Comments On Reviewer Discussion:**

Overall, reviewers were not enthusiastic about this work with two reviewers (H6Z8,3xqr) giving a rating of 3 (reject) and two reviewers (tM3P,dLVZ) giving a rating of 5 (marginally below acceptance).  Reviewer opinions where unchanged after author responses.

During the author response period, the authors provided additional results for Claude, Molmo, and a more recent version of GPT-4o.  Authors also investigated using different ways to prompt the VLM.  However reviewers remain unconvinced.

Overall, reviewers had questions about the positioning and motivation of the work. It was not clear to the reviewers whether the aim is
1) to assess 3D reasoning of recent visual-language models in which case there should be more effort in studying how to present the 3D information to the VLM [3xqr].
2) to assess spatial reasoning of 2D VLMs in which case reviewers felt the findings of this work didn't add to that of prior work (see comments by 3xqr, tM3P)

For studying of 3D reasoning, reviewers also recommend inclusion of 3D LLMs [3xqr,H6Z8,tM3P], or even 2D video models [3xqr].

---

### Decision · Program_Chairs · 2025-01-22

Reject